# Molten Sn solvent expands liquid metal catalysis

Junma Tang [1,2,3] ✉, Nastaran Meftahi[4], Andrew J. Christofferson [5,6], Jing Sun[7], Ruohan Yu[2], Md. Arifur Rahim [1,8], Jianbo Tang [2,9], Guangzhao Mao [2,10], Torben Daeneke [11], Richard B. Kaner [12,13], Salvy P. Russo [5,6] & Kourosh Kalantar-Zadeh [1,2] ✉

Regulating favorable assemblies of metallic atoms in the liquid state provides promise for catalyzing various chemical reactions. Expanding the selection of metallic solvents, especially those with unique properties and low cost, enables access to distinctive fluidic atomic structures on the surface of liquid alloys and offers economic feasibility. Here, Sn solvent, as a low-cost commodity, supports unique atomic assemblies at the interface of molten $SnIn_{0.1034}Cu_{0.0094}$, which are highly selective for $H_2$ synthesis from hydrocarbons. Atomistic simulations reveal that distinctive adsorption patterns with hexadecane can be established with Cu transiently reaching the interfacial layer, ensuring an energy-favorable route for $H_2$ generation. Experiments with a natural oil as feedstock underscore this approach's performance, producing $1.2 \times 10^{-4}$ mol/min of $H_2$ with 5.0 g of catalyst at ~93.0% selectivity while offering reliable scalability and durability at 260 °C. This work presents an alternative avenue of tuning fluidic atomic structures, broadening the applications of liquid metals.

Liquid metal catalysts offer opportunities and mechanistic insights beyond conventional catalysis[1–7]. In liquid metal catalysis, the mobility-induced fluidic structures of metallic atoms in molten alloys often lead to energy-favorable adsorption patterns with the reactants, originating from the alignments of their transient structures[8–10]. To date, Ga has been extensively used as a standard solvent in the development of liquid metal catalysts[11–14]. However, simply adjusting the compositions in Ga solvent cannot guarantee effective adsorptions with reactants that require specific atomic assemblies, as a result of the inherent behavior of the dissolved atoms within Ga. The high cost also restricts

the scaling up of Ga-based reaction systems[15]. In principle, the distribution of liquid atoms varies across different metallic solvents, which can result in distinct atomic assemblies. Therefore, expanding the choice of alternative solvents with low prices and varied physiochemical properties promises practical value and the potential to catalyze diverse reactions.

Molten Sn, with a much lower price than Ga[16], is capable of dissolving multiple elements and lowering the melting points of associated alloys to below 250 °C, thus holding potential as an ideal solvent[17–19]. Owing to the larger atomic radius (1.45 vs. 1.30 Å)[20,21],

[1]School of Chemical and Biomolecular Engineering, The University of Sydney, Sydney, Australia. [2]School of Chemical Engineering, University of New South Wales (UNSW), Sydney, Australia. [3]School of Chemistry, Xi'an Jiaotong University, Xi'an, China. [4]Department of Civil and Construction Engineering, Swinburne University of Technology, Melbourne, Australia. [5]School of Science, STEM College, RMIT University, Melbourne, Australia. [6]ARC Centre of Excellence in Exciton Science, School of Science, RMIT University, Melbourne, Australia. [7]Centre for Plasma Biomedicine, School of Electrical Engineering, Xi'an Jiaotong University, Xi'an, China. [8]Department of Chemical and Biological Engineering, Monash University, Clayton, Australia. [9]School of Engineering and Research Center for Industries of the Future, Westlake University, Hangzhou, China. [10]School of Engineering, Institute for Materials and Processes, The University of Edinburgh, Edinburgh, UK. [11]School of Engineering, RMIT University, Melbourne, Australia. [12]Department of Chemistry and Biochemistry and California NanoSystems Institute, University of California, Los Angeles, Los Angeles, USA. [13]Department of Materials Science and Engineering, University of California, Los Angeles, Los Angeles, USA. ✉e-mail: junma.tang@xjtu.edu.cn; kourosh.kalantarzadeh@sydney.edu.au

higher electronegativity (1.96 vs. 1.81)[22] and faster self-diffusion rate at catalytically relevant temperatures (3.35 vs. 1.17 × 10⁻⁵ cm²/s)[23] of molten Sn compared to liquid Ga, dissolved atoms in Sn solvent can exhibit different reactivity and distributions compared to their behavior in Ga solvent. The distinct surface properties of Sn and Ga may also impact the catalytic performance of their respective liquid alloys. For instance, although oxide layers spontaneously form on both Ga and Sn surfaces with negative Gibbs free energies, the oxide layer on the Sn surface does not fully passivate the interface, potentially exhibiting enhanced surface access for catalytic activities[24–26]. These fundamental differences between Sn and Ga can significantly alter the structures and, consequently, catalytic properties of dissolved atoms. Hence, by tuning the compositions in Sn solvent, distinctive structures of fluidic atoms could be established, possibly enabling characteristic adsorptions and facilitating reactions that are not feasible in other media.

As a proof of concept, we demonstrate a Sn-based molten alloy, $SnIn_{0.1034}Cu_{0.0094}$, that enables the selective synthesis of $H_2$ from various hydrocarbons at 260 °C. At the interface of molten $SnIn_{0.1034}Cu_{0.0094}$, In and Cu atoms remain fluidic, with In primarily located at positions of relatively low Sn density and Cu primarily embedded near the bottom of the interfacial layer. According to computational simulations, a Cu atom migrates and becomes transiently exposed at the interfacial layer, forming a unique structure with neighboring In and Sn atoms in the presence of hexadecane. This fluidic structure of metallic atoms establishes a distinctive adsorption pattern with hexadecane, ensuring an energy-favorable reaction pathway for $H_2$ generation with a selectivity of ~98.0%. Additionally, Cu atoms become more readily exposed with increasing In concentrations in molten Sn, thereby enhancing the efficiency of $H_2$ production. As a further demonstration, a renewable hydrocarbon model, canola oil, was employed as an alternative feedstock for extended applications of

this mechanism. At 260 °C, $H_2$ production with an efficiency of 1.2 × 10⁻⁴ mol/min and a selectivity of ~93.0% was obtained using 5.0 g of $SnIn_{0.1034}Cu_{0.0094}$ particles as catalysts. Investigating such fluidic atom structures in molten Sn solvent deepens our understanding of liquid metal catalysts and could broaden the selection of metallic solvents applicable to various other reactions.

## Results

### Sn-based molten alloy

A series of Sn-based alloys, including $SnIn_{0.0103}Cu_{0.0094}$, $SnIn_{0.0518}Cu_{0.0094}$, $SnIn_{0.1034}Cu_{0.0094}$ and $SnIn_{0.5169}Cu_{0.0094}$, were synthesized by dissolving In and Cu into molten Sn. Here, Sn serves as the metallic solvent while Cu is added to accept H directly from the hydrocarbon feedstock. In these combinations, the atomic ratio of In, which ensures the appearance of Cu at the interface were variables. Among them, $SnIn_{0.1034}Cu_{0.0094}$ was chosen for mechanistic investigations and further catalytic applications as a result of its better performance. Within the solubility limits, In and Cu remain atomically dispersed and fluidic at 260 °C with $SnIn_{0.1034}Cu_{0.0094}$ accessible in molten form (Fig. 1a)[18,27,28].

Ab initio molecular dynamics (AIMD) simulations were performed to investigate the status of liquid atoms in $SnIn_{0.1034}Cu_{0.0094}$ at 260 °C. At the interface of $SnIn_{0.1034}Cu_{0.0094}$, the closest contacts were found to be Cu–Sn and Cu–In (both 2.8 Å), followed by Sn–Sn (3.2 Å), In–In (3.3 Å), and Sn–In (3.3 Å), without any aggregations of In or Cu atoms (Fig. 1b). The atomic density profiles show that In atoms predominantly occupy positions of relatively low Sn density in the z dimension, while Cu atoms remain embedded within the interfacial layer (Fig. 1c). Additionally, throughout the 200 ps simulation period, no direct Cu-Cu contacts were observed (Fig. 1d), and Cu atoms frequently transitioned between the bulk and the interfacial layer in the molten Sn medium (Fig. 1e).

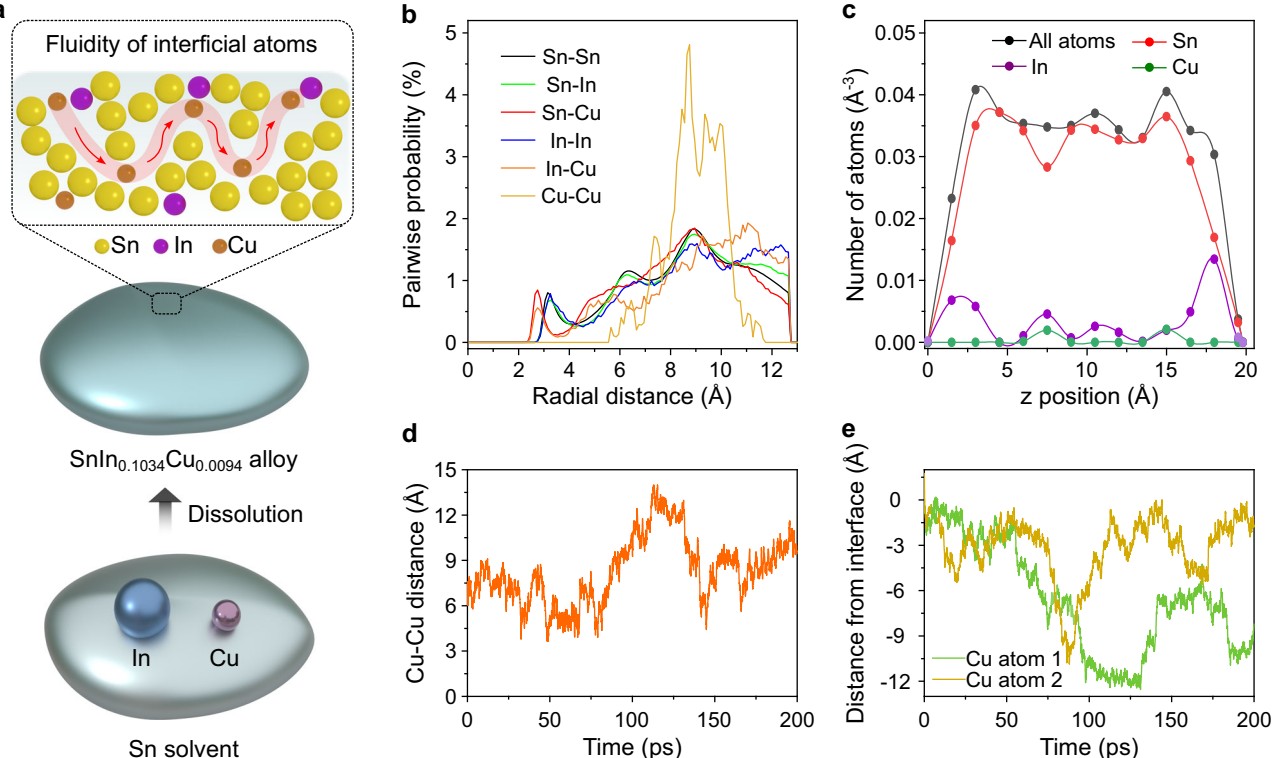

**Fig. 1 | Schematics and computational modeling of molten $SnIn_{0.1034}Cu_{0.0094}$.** **a** Schematics illustrating the preparation of $SnIn_{0.1034}Cu_{0.0094}$ and the fluidity of dissolved atoms. **b** Pairwise probability function for Sn, In, and Cu atoms at the interface of the molten alloy. **c** Atomic density profiles for all atoms, Sn, In, and Cu as a function of z position at the interfacial layer after 200 ps. **d** Distance between the two Cu atoms as a function of time. **e** Distance from the interface for the two Cu atoms, where the interface is defined as the midpoint of the interfacial layer based on the atomic density profile. Source data are provided as a Source Data file.

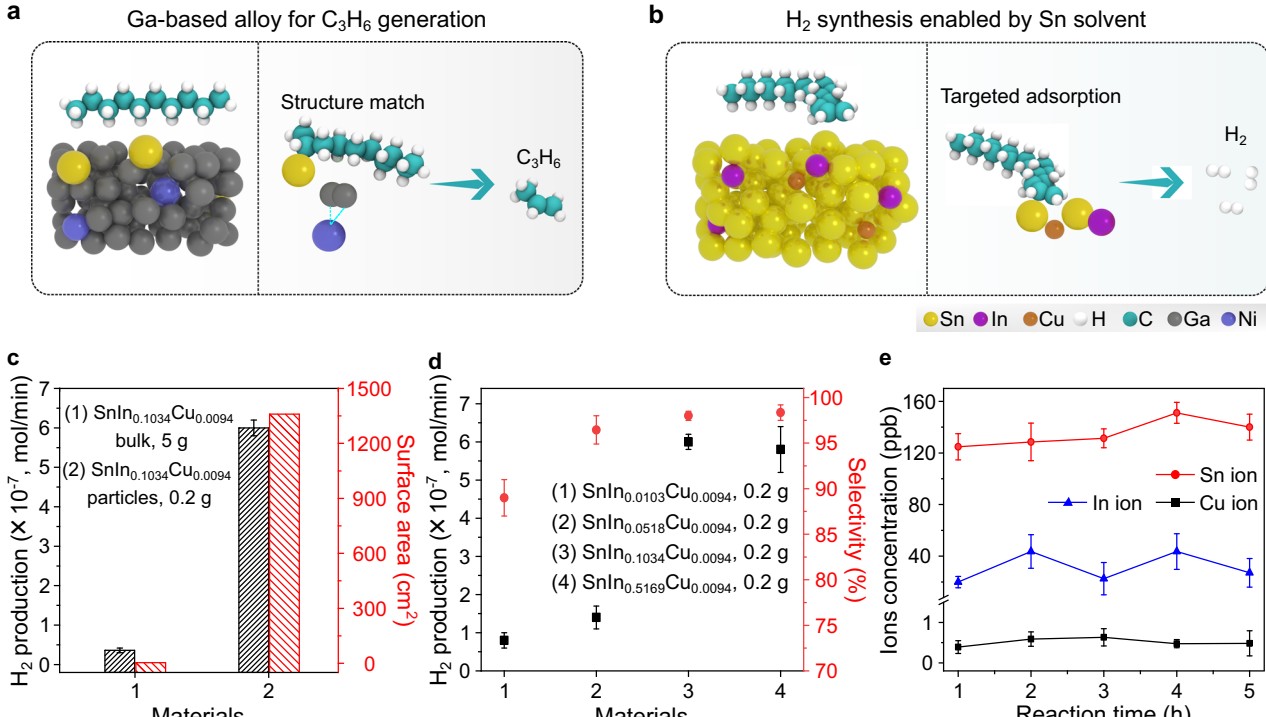

**Fig. 2 | Sn-based alloys for H₂ synthesis with hexadecane as feedstock.**
**a, b** Schematics depicting the generation of different products from hydrocarbons originating from the distinct atomic structures and the unique adsorption patterns on the surface of Ga (**a**) and Sn (**b**) based liquid alloys. **c** Efficiency and surface area of SnIn$_{0.1034}$Cu$_{0.0094}$ particles and bulk alloy for H₂ production. Data are presented as mean values ± SD, and $n = 2$ in each group. **d** Materials comparison of Sn-based alloys for H₂ production. Data are presented as mean values ± SD, and $n = 2$ in each group. **e** ICP-MS analysis of Sn, In and Cu ions in the liquid feedstock during the reaction. Data are presented as mean values ± SD, and $n = 2$ in each group. Source data are provided as a Source Data file.

## Table 1 | H₂ synthesis from hexadecane using different catalysts

| Catalysts[a] | Reaction temperature | Hydrocarbons[b] | Reaction time | H₂ (mol/min) | Ratio of H₂ |
|---|---|---|---|---|---|
| Sn | 260 °C | Hexadecane | 60 min | B.D.L.[c] | n/a |
| SnCu$_{0.0094}$ | 260 °C | Hexadecane | 60 min | B.D.L. | n/a |
| SnIn$_{0.1034}$ | 260 °C | Hexadecane | 60 min | B.D.L. | n/a |
| Oxidized SnIn$_{0.1034}$Cu$_{0.0094}$ | 260 °C | Hexadecane | 60 min | B.D.L. | n/a |
| SnIn$_{0.1034}$Cu$_{0.0094}$ | 260 °C | Hexadecane | 60 min | $3.6 \times 10^{-8}$ | ~98.0% |

[a]Reactions were performed using 5 g of bulk materials as the catalysts.
[b]The volume of hydrocarbon used for reaction was approximately 15 ml.
[c]B.D.L represents beyond detection limit. The detection limit of H₂ for the employed GC instrument is ~10 ppm.

Cyclic voltammetry was performed to explore the distributions of Cu and In atoms within molten Sn solvent and with reference to their surfaces. The results revealed that Cu atoms remained below the interfacial layer of molten Sn and could only reach the surface in the presence of In atoms (Supplementary Fig. 1, detailed discussions presented in Supplementary discussions). These observations are in accordance with the computational simulations.

### Fluidic atomic structure in molten Sn for H₂ synthesis
The selection of metallic solvents leads to discrete interfacial structures of the dissolved atoms and, consequently, distinctive catalytic phenomena. These include unique adsorption patterns with reactants and specific reaction pathways. For instance, with hydrocarbons as feedstock, the atomic configuration in Ga solvent facilitates the selective synthesis of propylene (Fig. 2a)[8]. In this work, molten Sn solvent provides an avenue for accessing a particular fluidic structure that enables energy-favorable adsorptions with hydrocarbons and selective H₂ generation (Fig. 2b).

Initially, SnIn$_{0.1034}$Cu$_{0.0094}$ particles (median diameter of 1200 nm) were prepared and loaded on glass microfiber filter papers to increase the surface-to-volume ratio of catalysts (Supplementary Figs. 2 and 3). At the reaction temperature of 260 °C, an H₂ production at a rate of $6.0 \times 10^{-7}$ mol/min and a selectivity of ~98.0 % was obtained using 0.2 g of SnIn$_{0.1034}$Cu$_{0.0094}$ particles as catalysts and hexadecane as feedstock, which was ~20 times higher than that of using bulk alloy (Fig. 2c, calculations presented in Supplementary discussions). Trace amounts of other gaseous byproducts, including CH₄, C₂H₄ and C₃H₆, were also detected, with no evidence of solid byproduct generation (Supplementary Fig. 4). Notably, due to the increased In concentration, SnIn$_{0.1034}$Cu$_{0.0094}$ exhibited enhanced efficiency and selectivity when compared with SnIn$_{0.0103}$Cu$_{0.0094}$ and SnIn$_{0.0518}$Cu$_{0.0094}$ (Fig. 2d and Supplementary Table 1). Despite the comparable performances of SnIn$_{0.1034}$Cu$_{0.0094}$ and SnIn$_{0.5169}$Cu$_{0.0094}$, the lower In content in SnIn$_{0.1034}$Cu$_{0.0094}$ offers a more economical option. Therefore, SnIn$_{0.1034}$Cu$_{0.0094}$ was considered the optimal catalyst for H₂ production.

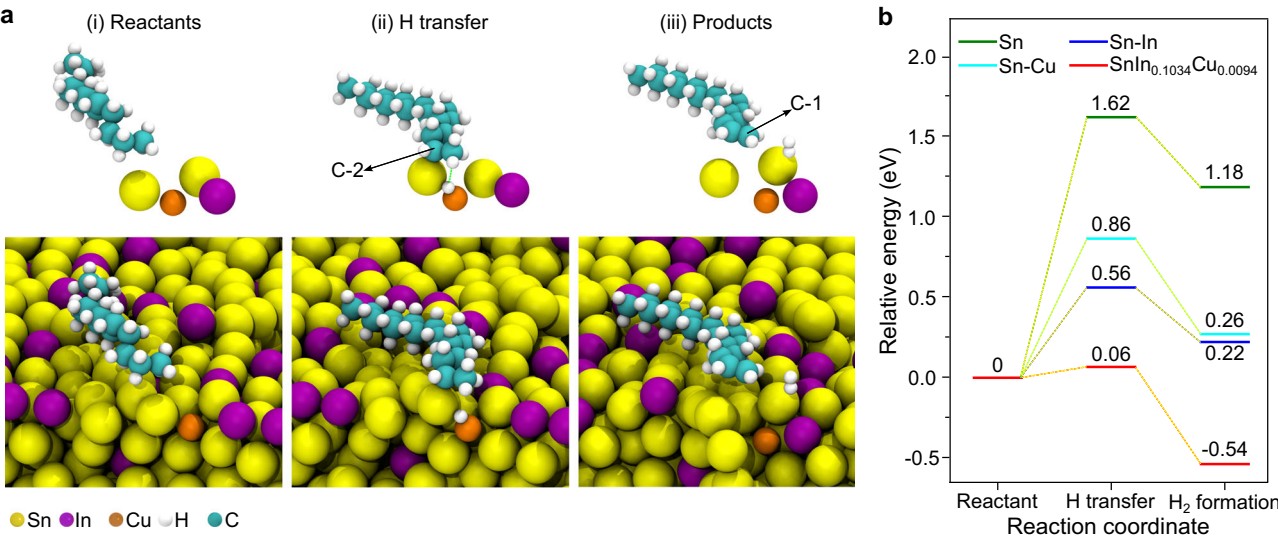

**Fig. 3 | Reaction mechanisms. a** Computational modeling of the reaction pathways for $H_2$ generation. The notation of C-1 and C-2 here denotes the locations of carbon atoms in hexadecane. **b** Energy barriers for $H_2$ synthesis from hexadecane by using different materials. Source data are provided as a Source Data file.

Inductively coupled plasma mass spectrometry (ICP-MS) analysis showed no obvious increase of Sn, In and Cu ions in the liquid feedstock during the reaction, indicating that $H_2$ generation occurs via a catalytic process without consuming the catalyst (Fig. 2e). Meanwhile, Fourier transform infrared spectroscopy and nuclear magnetic resonance analysis revealed the formation of C=C bonds in the hydrocarbon feedstock after the reaction, suggesting that $H_2$ formation originates from C-H bond dissociation reactions (Supplementary Figs. 5 and 6)[29,30].

Hydrogen generation was not feasible when using Sn, $SnIn_{0.1034}$, $SnCu_{0.0094}$ and oxidized $SnIn_{0.0103}Cu_{0.0094}$ as the catalysts, revealing that the synergy of In and Cu atoms, along with their fluidity, is a prerequisite for $H_2$ production (Table 1 and Supplementary Fig. 7). Additionally, no obvious $H_2$ generation was detected by conducting the experiments at the reaction temperature of 150 and 200 °C, which are below the melting point of $SnIn_{0.1034}Cu_{0.0094}$ alloy (Supplementary Table 2). This observation suggests that liquid state of $SnIn_{0.1034}Cu_{0.0094}$ alloy is the prerequisite for $H_2$ generation.

### Catalytic mechanisms

To gain mechanistic insights, AIMD simulations were performed to investigate the reaction pathways with hexadecane as the probe molecule. In the presence of hexadecane, the Cu atom migrates and transiently exposes itself at the interfacial layer. This exposure leads to the formation of a relatively high-energy atomic configuration, which subsequently establishes a unique adsorption pattern with hexadecane (Fig. 3a, i). Following adsorption, the Cu atom accepts an H from C-2, with C-2 being concurrently stabilized by a neighboring Sn atom (Fig. 3a, ii). The energy difference for this step is calculated to be 0.06 eV compared to the reactant configuration, and the energy penalty for removing an H from C-2 is compensated, to some extent, by the relaxation of the hexadecane conformation on the surface. Subsequently, the dissociated H interacts with an H on C-1, leading to the generation of $H_2$ and 1-hexadecene (Fig. 3a, iii). Upon the release of the generated species, the Cu atom sinks below the interfacial layer, resulting in a lower-energy surface configuration and an overall product configuration of −0.54 eV relative to reactants.

The overall process is energetically favorable, which is attributed to the fluidic behavior of Cu atoms. Given that Cu atoms would disperse from the interface in less than 2 ps in the absence of In atoms, the presence of In is crucial for ensuring Cu exposure. This phenomenon also explains the enhanced performance of Sn-based alloys with increased In concentrations. To confirm the role of Cu atoms in the reaction, a series of alloys with varying Cu concentrations, including $SnIn_{0.1034}Cu_{0.0047}$, $SnIn_{0.1034}Cu_{0.0018}$ and $SnIn_{0.1034}Cu_{0.0009}$, were synthesized. The reactivity and selectivity of these molten alloys for $H_2$ generation reduced as Cu concentration decreased, with $SnIn_{0.1034}Cu_{0.0018}$ and $SnIn_{0.1034}Cu_{0.0009}$ failing to produce $H_2$ from hexadecane (Supplementary Table 3). The lowered efficiency is likely due to the lesser availability of Cu atoms on the surface of the molten alloys. These findings indicate that the presence of surface Cu atoms and the interaction between In and Cu are crucial for the reaction.

Compared to $SnIn_{0.0103}Cu_{0.0094}$, the barriers for $H_2$ synthesis on other materials, including Sn, Sn-Cu and Sn-In, are energetically unfavorable, highlighting the synergistic effect of Cu and In atoms (Fig. 3b). As shown in Fig. 3b, the reaction profiles were recalculated with Cu atoms transformed to Sn (i.e., Sn and In only), with In atoms transformed to Sn (i.e., Sn and Cu atoms only), and with both In and Cu atoms transformed to Sn (i.e., Sn atoms only). In the Sn and In only system, the removal of H from C-2 changes the process from essentially energetically neutral in the presence of Cu to a process that is energetically unfavorable by 0.56 eV. While the product configuration is 0.34 eV lower in energy than the intermediate, it is still 0.22 eV higher in energy than the reactants. This is in line with the experimental results. In the absence of In (i.e., Sn-Cu and Sn only systems), the removal of H from C-2 rises to 0.86 eV for Sn-Cu and 1.62 eV for Sn only, in line with the experimental observations. These results highlight the synergistic role of In and Cu in stabilizing H on the surface and the dynamic role of Cu in facilitating an energetically favorable process overall. To further investigate the role of In, snapshots from AIMD simulations were taken where the Cu was exposed at the interface in the presence of hexadecane, alchemically transformed In to Sn, and continued the simulations. In all cases, the Cu diffused away from the interface in less than 2 ps, indicating that the role of In may be both stabilizing H on the surface and facilitating the presence of Cu at the interface.

Meanwhile, several possible atomic configurations at the interface were investigated for $H_2$ synthesis from hexadecane, including Cu activating Sn and/or In rather than interacting directly (Supplementary Fig. 8). In these configurations, when an H atom was removed from C-2 in hexadecane and adsorbed on the surface, it either spontaneously returned to the C-2 during AIMD simulations or resulted in

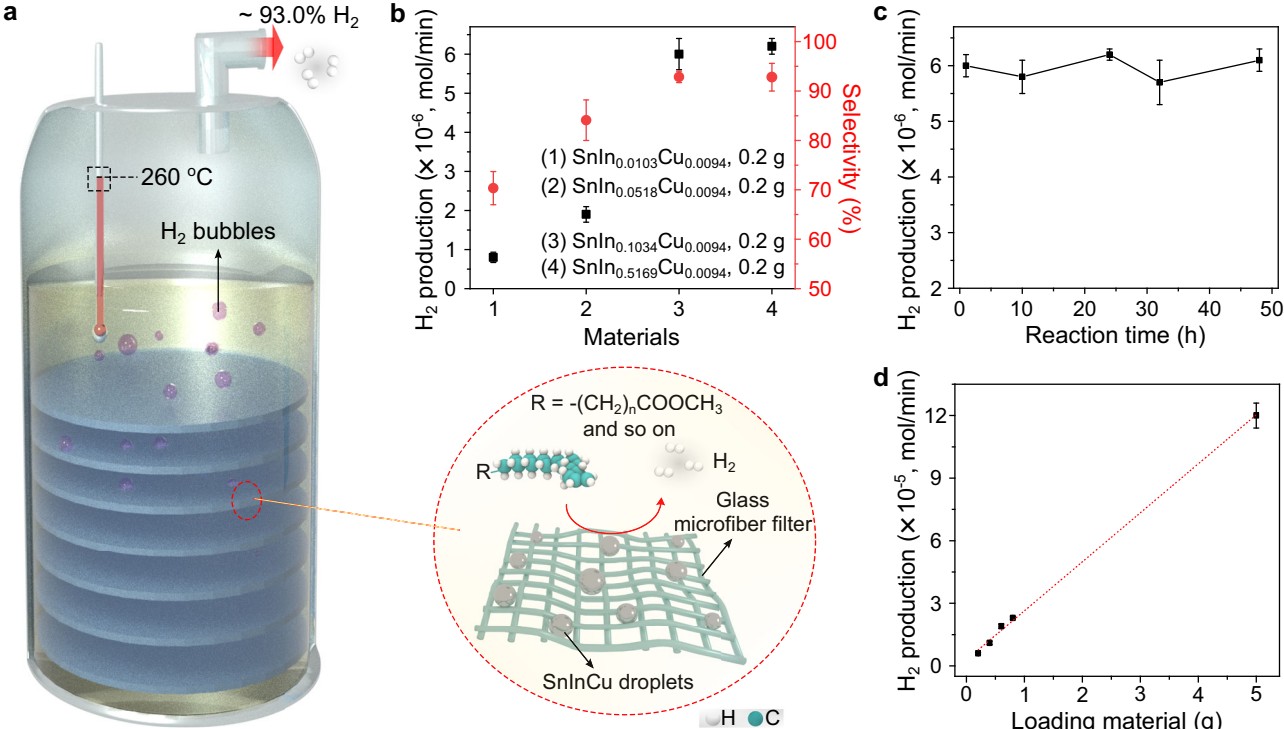

**Fig. 4 | Schematics and experimental results using canola oil as feedstock.**
**a** Schematics demonstrating the reaction system with $SnIn_{0.0103}Cu_{0.0094}$ particles loaded on glass microfiber filter papers as catalysts and canola oil as feedstock. The inset (dashed red circle) illustrates a zoomed-in view. **b** Comparison of Sn-based alloys for $H_2$ production using canola oil as feedstock. Data are presented as mean values ± SD, and $n$ = 2 in each group. **c** Long-term experiment for over 48 h. Data are presented as mean values ± SD, and $n$ = 2 in each group. **d** Scaled-up experiments. Data are presented as mean values ± SD, and $n$ = 2 in each group. Source data are provided as a Source Data file.

configurations that were energetically unfavorable by ~2 eV. The catalytic process becomes energy-favorable only when Cu is exposed on the interfacial layer and directly interacts with hexadecane.

Originating from the dynamicity and mobility of liquid atoms, AIMD simulations face limitations for molten alloys. The simulation boxes of liquid metallic atoms are small and periodic, and the timescales are short. Moreover, the high computational cost of modeling liquid hexadecane interfaced with liquid metal restricts the simulation to one single hexadecane molecule. To gain a more comprehensive understanding of this reaction, the Arrhenius equation was employed to investigate the activation energy. An Arrhenius plot was generated by performing experiments at varying reaction temperatures (Supplementary Fig. 9 and Supplementary Table 4, detailed calculations provided in the Supplementary discussions). The activation energy was calculated to be approximately 1.10 eV, which is lower than that of other simulated reaction pathways requiring at least ~2 eV (Supplementary Fig. 8). These findings offer additional insight into the activation energy, further reinforcing the proposed reaction mechanisms.

Simulations also show that the distinct atomic structures in Ga and Sn-based alloys are responsible for the different reaction pathways of hydrocarbons (Supplementary Fig. 10, a detailed discussion is provided in the Supplementary section). All these simulations correlate to our experimental results, validating the proposed reaction mechanisms.

### Extended applications for $H_2$ generation

To broaden the applications of this reaction mechanism, a renewable hydrocarbon, canola oil, was further employed as a proof-of-concept feedstock[31,32]. Glass microfiber filter papers loaded with $SnIn_{0.1034}Cu_{0.0094}$ particles were stacked in the reactor (Fig. 4a). At 260 °C, an $H_2$ production rate of $6.0 \times 10^{-6}$ mol/min with a selectivity of

~93.0% was obtained by using 0.2 g of $SnIn_{0.1034}Cu_{0.0094}$ particles as the catalyst. The efficiency is ~10 times higher than that achieved using hexadecane as feedstock. Comparable results to the hexadecane case were observed in terms of catalyst composition comparisons, and $SnIn_{0.1034}Cu_{0.0094}$ demonstrated better performance than other alloys (Fig. 4b and Supplementary Table 5). The durability of this reaction system was demonstrated through a continuous 48-h reaction (Fig. 4c and Supplementary Table 6). The turnover number for $H_2$ synthesis based on the long-term experiment was estimated to be ~3.4 × 10^5 (Calculations presented in the Supplementary section). Meanwhile, there was no noticeable change in the size of the $SnIn_{0.1034}Cu_{0.0094}$ particles after the reaction, suggesting that the molten droplets remained separate and did not aggregate during the process (Supplementary Fig. 11). For comparison, the $SnIn_{0.1034}Cu_{0.0094}$ particles of different diameters were also synthesized (Supplementary Table 7). The exposed surface area of $SnIn_{0.1034}Cu_{0.0094}$ increased as the particle size decreased, and a reduction in particle size resulted in increased $H_2$ generation rates (Supplementary Fig. 12). These results reveal that the exposed surface area of the catalysts is a key factor in determining the reaction system's $H_2$ production rate.

This setup can be readily scaled up by loading more catalysts, with a linear increase in $H_2$ generation observed when using 0.4, 0.6, 0.8 and 5.0 g of $SnIn_{0.1034}Cu_{0.0094}$ particles as the catalyst (Fig. 4d, Supplementary Table 8 and Supplementary Video 1). Post-reaction system characterizations revealed the emergence of C=C bonds in the hydrocarbon feedstock without evidence of solid byproduct generation (Supplementary Figs. 13–15)[29,30]. These observations suggest that canola oil likely undergoes the same reaction pathways as hexadecane on the surface of $SnIn_{0.1034}Cu_{0.0094}$, validating the broad applicability of this catalyst and corresponding catalytic mechanisms. To investigate the reason for enhanced efficiency of $H_2$ generation in canola oil

case, oleic acid was further selected as the feedstock owing to some structural similarity to canola oil. Under the same reaction conditions, an increased $H_2$ generation rate of $1.8 \times 10^{-5}$ mol/min, with the selectivity of ~84.5%, was obtained by using oleic acid hydrocarbon source (Supplementary Table 9). The increased reaction rate is likely attributed to the presence of the carboxyl group in oleic acid, which partially explains the higher efficiency observed with canola oil, given their structural similarity.

Additionally, comparing with other reported materials, molten $SnIn_{0.1034}Cu_{0.0094}$ alloy exhibits several advantages for $H_2$ generation from hydrocarbons, including low cost of catalyst preparations (~US\$0.7/$g_{catalyst}$) (reference: stock price Oct. 2024), mild operating conditions and the use of renewable feedstocks (Supplementary Table 10, detailed discussions presented in Supplementary discussions).

## Discussion

In conclusion, the interfacial structures of fluidic atoms in molten alloys can be tuned through the selection of solvents, thereby demonstrating unique catalytic behaviors. Utilizing molten Sn as a low-cost metallic solvent at a moderate temperature, transient structures containing Sn, In and Cu atoms are formed at the interface of $SnIn_{0.1034}Cu_{0.0094}$. In contrast to Ga-based alloys, the distinctive atomic structures in Sn solvent facilitate unique adsorptions and selective reaction pathways for $H_2$ synthesis. Hydrocarbons, including hexadecane and canola oil, were efficiently converted into $H_2$ at the interface of molten $SnIn_{0.1034}Cu_{0.0094}$ with high selectivity at 260 °C. This work deepens our understanding of liquid metal catalysts. Exploring cost-effective metallic solvents offers a viable strategy to tailor the catalytic properties of molten metals and encourages further exploration of liquid metal catalysts for various other reactions.

## Methods
### Materials
Tin (Sn, purity: 99.9%), indium (In, purity: 99.9%) and copper film (Cu, purity: 99.9%) were obtained from Sigma-Aldrich. Hexadecane (purity: ≤99%) and glass microfiber filter (100 circles, diameter 47 mm) were purchased from Sigma-Aldrich. Canola oil (mainly composed of triacylglycerols with 6–14% α-linolenic acid, 50–66% oleic acid and <7% of saturated fatty acids) was acquired from a local consumable product supermarket. HCl (33 wt% in water) was acquired from Chem-Supply Pty Ltd. Milli-Q water was used throughout the experiments for sample preparations. $CDCl_3$ (purity: <99.9%) was purchased from Sigma-Aldrich.

### Sample characterizations
XPS analysis was performed on a Thermo Scientific K-alpha X-ray spectrometer. The liquid source was studied using micro-FTIR spectroscopy on a PerkinElmer Spectrum 100 FTIR Spectrometer which is coupled to a Spotlight 400 FTIR Imaging System with stage controller. The morphology and structure of materials were imaged by SEM (JEOL JSM-IT-500 HR) with an EDS detector for elemental and compositional analysis. The ICP-MS experiment was performed on a NexION 2000 B ICP Mass Spectrometer to determine the concentration of Sn, In and Cu ions. NMR experiments were performed to investigate the liquid species in the hydrocarbon sources, which was carried out by using a Bruker Avance III 600 MHz Cryo NMR (Ernst).

### MD simulations
Initial classical molecular dynamics (MD) simulations were performed with 200 Sn atoms in a $17.825 \times 17.825 \times 17.825$ Å³ box using the MD code LAMMPS[33] in order to rapidly generate equilibrated configurations of the liquid metal. Force field parameters for Sn were taken from our previous work[19]. Following this initial equilibration, 21 Sn atoms were alchemically converted to In and two Sn were alchemically converted to Cu to give the experimental ratio of 177 Sn: 21 In: 2 Cu (10 wt% In, 0.5 wt% Cu). The In atoms were randomly distributed throughout the system, while the two Cu atoms were initially placed ~8 Å apart at the interface. A 10 Å vacuum spacer was added in the z dimension and interfacial ab initio MD (AIMD) simulations were performed on this system for 200 ps with a 4 fs timestep using the Vienna ab initio Simulation Package (VASP)[34,35] at 533.15 K with the projector-augmented wave (PAW)[36] method, the PBE exchange correlation functional[37], an energy cutoff of 420 eV, and the gamma point only for the k-point grid. For simulations involving hexadecane, random initial configurations of hexadecane were added to snapshots of the liquid metal interface following 200 ps of AIMD simulation where at least one Cu was present in the interfacial layer. The timestep was reduced to 0.5, and the vacuum spacer was extended to 15 Å. To generate the reaction profile, multiple configurations were examined dynamically in a procedure similar to that outlined by Ruffman et al.[38]. Geometry optimizations were performed on AIMD snapshots for each step in the reaction with a $4 \times 4 \times 1$ k-point grid. All other analyses were performed using VMD 1.9.3[39,40].

The DFT-D3 method with Becke-Johnson damping function (IVDW = 12 in VASP) dispersion correction was used for all simulations and geometry optimizations. All simulations were performed in the NVT ensemble with the temperature controlled by the Nose–Hover thermostat.

### Materials preparation
After a prewash using a 0.1 M solution of HCl to eliminate the oxide layers, 1 g of In, 0.05 g of Cu and 10 g of Sn were mixed to synthesize $SnIn_{0.1034}Cu_{0.0094}$ alloy. The solubility limit of Cu in molten Sn solvent is ~1 wt% at 260 °C. The mixtures were heated at ~400 °C inside a $N_2$-filled glove box for a few hours until In and Cu were completely dissolved in Sn solvent.

To synthesize the particles, 5 g of bulk $SnIn_{0.1034}Cu_{0.0094}$ alloy was initially placed into a glass vial containing 10 ml of glycerol. This mixture was subjected to ultrasonic vibrations using a probe sonicator (model VC 750 from Sonics & Materials) under the protection of $N_2$ while being heated to ~300 °C to keep the alloy molten. The sonication amplitude was set to 55%, corresponding to an ultrasonic power input of ~410 W. The sonicator was set to pause for 1 s after each 9 s sonication and the total sonication time was 30 min. After the sonication process, the particles were washed by using ethanol and Milli-Q water several times to thoroughly remove glycerol.

The preparation of $SnIn_{0.5169}Cu_{0.0094}$, $SnIn_{0.0518}Cu_{0.0094}$ and $SnIn_{0.0103}Cu_{0.0094}$ followed the same procedure as that of $SnIn_{0.1034}Cu_{0.0094}$.

### Experimental procedures

(1) Using bulk $SnIn_{0.1034}Cu_{0.0094}$ as catalyst: A 5.0 g droplet of $SnIn_{0.1034}Cu_{0.0094}$ was directly introduced into a reactor prefilled with 15 ml of hexadecane. The reaction setup was subsequently heated to and stabilized at ~260 °C. Argon gas was purged through the reaction system at the rate of ~13 sccm to recover the produced gaseous species for GC analysis. Control experiments employing Sn, SnIn, and SnCu as catalysts were conducted following the same procedure as that of the $SnIn_{0.1034}Cu_{0.0094}$ experiment to ensure comparability of results.

(2) Loading $SnIn_{0.1034}Cu_{0.0094}$ particles on the glass microfiber filter papers for $H_2$ production: $SnIn_{0.1034}Cu_{0.0094}$ particles (0.2 g) were dispersed in ethanol and subsequently drop-cast onto glass microfiber filter papers. The catalytic materials were washed with 0.1 M HCl solution to remove the oxide layer, vacuum-dried, and subsequently stacked within the liquid feedstock. The experiments were conducted at a temperature of ~260 °C. The hydrogen production efficiency was evaluated

by pumping argon gas through the system at a rate of ~13 sccm combined with GC analysis.

The experiments of using $SnIn_{0.5169}Cu_{0.0094}$, $SnIn_{0.0518}Cu_{0.0094}$ and $SnIn_{0.0103}Cu_{0.0094}$ particles as the catalysts followed the same procedure as that of $SnIn_{0.1034}Cu_{0.0094}$.

(3) Scaled-up experiments: Different amounts of $SnIn_{0.1034}Cu_{0.0094}$ particles were loaded on glass microfiber filter papers, which were subsequently dried and stacked in a reactor containing 50 ml of canola oil. The reactor was heated to a temperature of ~260 °C. Argon gas was flowed through the system at ~13 sccm to collect the gaseous products generated during the reaction for GC analysis.

(4) Long-term experiment: $SnIn_{0.1034}Cu_{0.0094}$ particles (0.2 g) were used as the catalyst with canola oil as the feedstock. During the reaction, argon gas at the flow rate of ~13 sccm was pumped continuously through the reaction system to recover the produced gaseous species. The experiment was allowed to run continuously for over 48 h at ~260 °C, and the gas samples for GC analysis were taken every ~10 h.

## ICP-MS analysis

To detect Sn, In, and Cu ions in the hexadecane solution throughout the reaction, 1.0 ml of the liquid hydrocarbon were extracted from a glass vial hourly during the conversion process. These samples were then evaporated completely by heating them on a hot plate at ~300 °C. Following evaporation, 1.0 ml of nitric acid was introduced into the glass vial to dissolve any remaining residues, preparing them for ICP-MS analysis.

## Data availability

The data generated in this study are provided in the Supplementary Information/Source Data file. Source data are provided with this paper.

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

## Acknowledgements

This work was supported by the Australian Research Council (ARC) Laureate Fellowship grant (FL180100053). This work was supported by computational resources provided by the Australian Government through the National Computational Infrastructure (NCI) and Pawsey Supercomputing Research Centre under the National Computational Merit Allocation Scheme.

## Author contributions

J.T. (Junma Tang) initiated the concept and designed the experiments along with K.K.-Z. J.T. (Junma Tang) conducted the experiments and carried out the characterizations with the help of K.K.-Z. The molecular dynamic simulations were performed by N.M., A.J.C., and S.P.R. The following individuals contributed to the data analyses, scientific discussions and authorship of the paper: A.J.C., N.M., J.S., R.Y., M.A.R., J.T. (Jianbo Tang), G.M., T.D., S.P.R., R.B.K., K.K.-Z. All authors revised the manuscript and provided helpful comments.

## Competing interests

The authors declare no competing interests.
