## [Peer Review File · Nature Communications]

Molten Sn solvent expands liquid metal catalysis

Corresponding Author: Professor Kourosh Kalantar-Zadeh

Version 0:

Reviewer comments:

Reviewer #1

(Remarks to the Author)

Molten Sn was used to produce reactive copper atom-dispersed fluidic surface, which was active for producing hydrogen gas from inert saturated hydrocarbons at 260 Celsius degrees with 93% selectivity. I enjoyed reading the manuscript. The data support the authors' arguments. Hence, I'd love to recommend publication of this work to the journal. One minor comment: discussion on the role of In should be moved from the Supplementary Information to the main texts.

Reviewer #2

(Remarks to the Author)

This manuscript describes a molten SnInCu catalysts for the conversion of canola oil and hexadecane into dehydrogenated products. The molten catalyst adds to the rapidly growing area of new molten catalysts and is a welcome addition. The authors suggest that the catalyst could be used for other more useful reactions, although these are not tested or simulated.

Supplementary Table 1 was the most noteworthy result. In it, the authors demonstrated that Sn, SnCu, and SnIn were inactive (below detection limit); however, SnInCu was active for hexadecane dehydrogenation. It should be noted that the detection limit should be given to fully support this! Could the table or parts of it be moved to the main text?

Fig 3 b was also among the most noteworthy results, and supports Table 1. However, other work on molten metals suggests that adsorbates and transitions states can fluctuate by as much as 1 eV over time, for a given composition. So a single configuration is less telling and other configurations should be considered, especially in light of the fact that this is one of the most noteworthy results.

The performance was quantified for their reactor, although the rate is not particularly high – although some comparison with other catalysts would be appreciated. The rate was give in units of mol/min for a give catalyst mass; although for their unique reactor, it is unclear if these units are best for comparison to other catalysts. Also, the conversion over the experiments should be specified. A back of the envelope calculation suggests less than 0.1%; however, this should be reported.

The manuscript also mentions the scalability of the catalytic process but does not provide detailed quantitative analysis or comparison with other systems. This lack of detailed scalability metrics (like catalyst loading, reaction volume scaling, and economic feasibility at larger scales) may leave the reader questioning the practical applicability of the findings.

The manuscript discusses AIMD simulations that probe the role of Cu and it's "fluidity". It seems that more could be done to understand why In is so important, based on the results from Table 1. For example, in line Lines 160-162: the explanation uses models which is Sn-In, Sn-Cu, and Sn only. To probe the role of In, the concentration of In should be changed to determine how that impacts the fluidity of Cu. They say In facilitates Cu's presence at the surface as Cu diffuses away in <2ps after In is transformed to Sn. What about the concentration dependence of In? It seems like more could be done to explain the synergy of In and Cu in the Sn solvent via AIMD simulations similar to what has already been done. For example: change only some of the In atoms to Sn and see how the system changes to investigate different In concentrations.

A few specific minor comments:

- Fig 1c: why is Cu not included at all positions? (if 0, include as 0)
- Figure 2a relates more to the group's other work than the present manuscript.

- Line 66-67: justify this statement with the specific observation that they authors have made to make the statement
- Lines 62-94 has some repetitions
- Lines 116-118: Supplementary Fig. 3 does not probe carbon via EDX, only Sn, In, Cu.
- The solubility limits of Cu in the melts should be given somewhere

Reviewer #3

(Remarks to the Author)

In this manuscript, the authors used molten Sn as a metallic solvent to dissolve multiple elements, thereby lowering the melting points of the resulting alloys. It was demonstrated that the alloy $\text{SnIn}_{0.1034}\text{Cu}_{0.0094}$ selectively synthesizes hydrogen (H_2) at the interface when reacting with various hydrocarbons, including hexadecane and canola oil, at 260°C. Furthermore, an ab initio molecular dynamics (AIMD) simulation study revealed that Cu atom migrate and transiently expose itself at the interfacial layer. Following the adsorption of hexadecane, a Cu atom accepts a hydrogen atom, with a neighboring Sn atom stabilizing the carbon site. Concurrently, fluidic In atoms play a crucial role in ensuring Cu exposure, enhancing H_2 production in Sn-based alloys as the In concentration increases. Similarly, hydrogen production from canola oil was investigated, revealing identical reaction pathways as those observed with hexadecane.

The paper proposes expanding the use of liquid metal catalysts by exploring Sn-based alloys as a cost-effective alternative to Ga-based alloys for practical applications. However, it currently lacks sufficient experimental details and descriptions for the observed catalytic trends and does not provide adequate data to support the proposed catalytic mechanisms. This affects the reliability of the findings. Additional experimental and analytical data, including those specified below, should be provided to strengthen the paper before it can be considered for publication in Nature Communications.

1. In this manuscript, $\text{SnIn}_{0.1034}\text{Cu}_{0.0094}$ catalyst appears to require more harsh condition for the H_2 synthesis compared to $\text{GaSn}_{0.029}\text{Ni}_{0.023}$. Including data on the variation in H_2 production of $\text{SnIn}_{0.1034}\text{Cu}_{0.0094}$ at different reaction temperatures would be advantageous. Does catalytic performance of $\text{SnIn}_{0.1034}\text{Cu}_{0.0094}$ decreased dramatically at temperatures lower than 260°C?

2. In Figure 2d and 4b, the systems with higher In content, specifically $\text{SnIn}_{0.1034}\text{Cu}_{0.0094}$ and $\text{SnIn}_{0.5169}\text{Cu}_{0.0094}$, demonstrate both high selectivity and significant H_2 production. The authors should provide a detailed explanation from the perspective of "catalytic selectivity". This should include an analysis of how Cu and In contribute to the observed catalytic behavior, particularly regarding the selective synthesis of H_2 and the overall catalytic efficiency.

3. The author prepared the Sn-based alloys with varying In ratios, including $\text{SnIn}_x\text{Cu}_{0.0094}$ ($x=0.0103, 0.0518, 0.1034$). In AIMD simulations, the proposed catalytic mechanisms suggests that the Cu and Sn serve as the active catalytic sites, with Cu adsorbing hydrogen (H) and Sn adsorbing carbon (C). Therefore, controlling the amount of Cu in the alloys could provide strong evidence supporting the observed catalytic tendencies and mechanisms.

4. The catalyst droplets with an average diameter of 1200 nm were used in the H_2 production tests. If the size of these droplets can be controlled, the effect of droplet size on catalytic performance should be described. Additionally, providing size distribution data of the droplets after the catalytic measurements is necessary to confirm whether aggregation occurred during the catalytic reactions. This information is crucial for understanding the impact of droplet size on catalytic efficiency and stability, as well as for optimizing the catalyst design.

5. The manuscript states that In is found in regions with relatively low tin Sn density, while Cu is primarily located near the bottom of the interfacial layer, occasionally becoming transiently exposed at the interface in the presence of hexadecane. However, the SEM images in the supplementary information do not adequately support this explanation. Additional data, such as high-resolution imaging or compositional analysis, is needed to substantiate these claims and provide a clearer understanding of the spatial distribution and interaction of In and Cu within the catalyst.

6. The energy barriers for H_2 synthesis are presented in Figure 3b, illustrating the hydrogen transfer and H_2 formation energies. However, the figure lacks descriptors for the adsorption energies of the reactants. The adsorption step is crucial for determining the overall reaction kinetics, so the authors should provide some description of the adsorption rate.

7. The authors noted that using canola oil as a feedstock resulted in an efficiency approximately ~10 times higher than when using hexadecane. However, the manuscript does not provide an explanation for this phenomenon. Since canola oil was mostly composed of fatty acids, the cause of this efficiency improvement may be due to fatty acids. Therefore, the authors should be able to reveal this enhancement by using some fatty acids (e.g., oleic acid or α -linolenic acid) as feedstock in this reaction.

Reviewer #4

(Remarks to the Author)

The authors have performed experimental and theoretical studies to explore the reactivity of a Sn-based ternary alloy at low temperatures. For this, they have used natural oil feedstock for producing H_2 . It is not clear to me what is the novelty of this work. Several works are showing Sn as a solvent liquid for active metals for the catalysis of hydrocarbons to H_2 . The authors may want to look at the works of Ogino et al. from the 1970s to 1980s or more recent works from the McFarland group at

UCSB.

The main conclusions of their paper have been drawn from simulations. However, I have serious concerns over the methodology used in the paper. Some of them are listed below:

1. It is difficult to understand how reaction barriers are obtained. The paper cited by authors to get reaction barriers only discusses getting dispersion in adsorption energies. The surface of the liquid is ever-changing. The authors also point this out and is the main conclusion of their paper. However, they report barriers based on one snapshot. Are the fluctuations or dynamic configurations not important? As per their main conclusions, dynamicity plays a major role. I would suggest authors perform more thorough free energy calculations using rare-event methods before this publication can be accepted.
2. The authors have not taken dispersions in their simulations. It is well known this could lead to significant errors.
3. It is unclear what ensemble they are working in and how the equilibration is performed. The authors mention AIMD simulation at 260 deg C. How are the authors controlling the temperature?

Version 1:

Reviewer comments:

Reviewer #1

(Remarks to the Author)

The authors have addressed the issues raised in the previous round of revisions. I have one final minor comment: on p.3, line 52, the authors compare molten Sn with Ga, and I believe a discussion of the surface properties of these two materials would be beneficial, as it is relevant to potential applications (such as catalysis). For instance, Ga forms a self-passivating Ga_2O_3 layer spontaneously with a negative Gibbs free energy (refs: ACS Appl. Mater. Interfaces 2018, 10, 40, 34758–34764; Small, 2020, 16, 12, 1903391). In contrast, while Sn also spontaneously forms oxides (SnO or SnO_2) with negative Gibbs free energies, the oxide layer on tin does not fully passivate the surface. This non-passivating behavior arises from the properties of the tin oxide layer, which is often porous and lacks sufficient adherence to prevent further oxidation. As a result, oxidation continues gradually, particularly under conditions of high humidity or elevated temperatures (ref: Progress in Surface Science, 2005, 79, 2-4, 47-154). Including a discussion on this distinction, with relevant citations, would add depth to the comparison. Once again, this is an excellent study, and I enjoyed reading it!

Reviewer #2

(Remarks to the Author)

I have only one further comment related to the authors responses.

While the authors added valuable information in Table R2, it would be unfair to list 9 catalysts that have rates (“efficiencies”) per g_catalyst, but then list their SnInCu catalyst with a rate per g_surface atoms. Is that how the values are calculated? If so, I suggest revising to have an apples-to-apples comparison (i.e. per g_catalyst in all cases or change the rate units all to be per square meter). As it stands, one cannot compare SnInCu to any of the other catalysts. Also, the comparison would be significantly improved if made to other catalysts for canola oil dehydrogenation, instead of other feedstocks.

Using the values in the new Table R1 of 3.6×10^{-8} (mol H₂/min) divided by 5 grams as listed in the table legend, one gets 7×10^{-9} mol/min/g_cat, NOT the value listed in Table R2. The value from Table R2 is also different from what is reported in the revised abstract. 7×10^{-9} mol/min/g_cat is many orders of magnitude lower than all catalysts listed in Table R2. While this isn't all that surprising since the other rates are at higher temperatures for alkanes, it should nonetheless be reported and calculated in a fair way.

Supplemental information should also include all of the information used to perform the calculations in the table (e.g. inlet flows, outlet flows detected, reactor volume, catalyst mass, surface area, etc.) so that reproduction of the values is possible.

Reviewer #3

(Remarks to the Author)

The work can be published.

Reviewer #4

(Remarks to the Author)

The envisaged corrections have been made and mostly the comments have been addressed. In the reply, the authors clearly acknowledge the shortcomings of the computational model. However, this has not been added to the main text of the manuscript. I will be very supportive of this work if these are added to the main text along with the effects they can have on the conclusions of this work.

Reviewer 1:

Molten Sn was used to produce reactive copper atom-dispersed fluidic surface, which was active for producing hydrogen gas from inert saturated hydrocarbons at 260 Celsius degrees with 93% selectivity. I enjoyed reading the manuscript. The data support the authors' arguments. Hence, I'd love to recommend publication of this work to the journal.

Re: We really appreciate for your very supportive comments.

One minor comment: discussion on the role of In should be moved from the Supplementary Information to the main texts.

Re: Thanks. To address your comment, the discussions regarding the role of In and Cu, presented in the Supplementary Information, are now moved to main manuscript, with some minor edits, on page 10 : “As shown in Fig. 3b, the reaction profiles were recalculated with Cu atoms transformed to Sn (*i.e.*, Sn and In only), with In atoms transformed to Sn (*i.e.*, Sn and Cu atoms only), and with both In and Cu atoms transformed to Sn (*i.e.*, Sn atoms only). In the Sn and In only system, the removal of H from C-2 changes the process from essentially energetically neutral in the presence of Cu to a process that is energetically unfavorable by 0.56 eV. While the product configuration is 0.34 eV lower in energy than the intermediate, it is still 0.22 eV higher in energy than the reactants. This is in line with the experimental results. In the absence of In (*i.e.*, Sn-Cu and Sn only systems), the removal of H from C-2 rises to 0.86 eV for Sn-Cu and 1.62 eV for Sn only, in line with the experimental observations. These results highlight the synergistic role of In and Cu in stabilizing H on the surface, and the dynamic role of Cu in facilitating an energetically favorable process overall. To further investigate the role of In, snapshots from AIMD simulations were taken where the Cu was exposed at the interface in the presence of hexadecane, alchemically transformed In to Sn, and continued the simulations. In all cases, the Cu diffused away from the interface in less than 2 ps, indicating that the role of In may be both stabilizing H on the surface and facilitating the presence of Cu at the interface.”

Reviewer 2:

This manuscript describes a molten SnInCu catalysts for the conversion of canola oil and hexadecane into dehydrogenated products. The molten catalyst adds to the rapidly growing area of new molten catalysts and is a welcome addition.

Re: We really appreciate for your supportive comments.

The authors suggest that the catalyst could be used for other more useful reactions, although these are not tested or simulated.

Re: Thanks. Indeed, liquid metal catalysis is an emerging field. So far, Ga has been shown in multiple catalytic applications, such as growth of diamond (*Nature*, 629, 348–354, 2024), dechlorination of polyvinyl chloride (*Sci. Adv.*, 10, eadm9963, 2024) and formate electrosynthesis (*Adv. Funct. Mater.*, 2408966, 2024).

However, the relatively high price of Ga limits its practical implementations for scaling up. Exploring alternative metallic solvents, with lower costs and distinct physiochemical properties, should logically lead to new avenues for diverse catalytic applications of liquid metals.

In our work, as a proof-of-concept, molten Sn was used as the alternative low-cost solvent. In comparison, the different atomic structures in Ga and Sn solvents reveal distinct catalytic behaviour with hydrocarbons as feedstock. We believe that this work can encourage the explorations of different metallic solvents for a variety of useful chemical reactions.

To further clarify the above point and address your comment, the sentence “Investigating such fluidic atom structures in molten Sn solvent deepens our understanding of liquid metal catalysts and broadens their applicability across a wider range of reactions.” in the introduction part on page 4 was revised to “Investigating such fluidic atom structures in molten Sn solvent deepens our understanding of liquid metal catalysts and could broaden the selection of metallic solvents applicable to various other reactions.”. Also, the sentence in conclusion part “Exploring cost-effective metallic solvents provides a valid strategy to tailor the catalytic properties of molten metals for potential practical applications.” was further revised into “Exploring cost-effective

metallic solvents offers a viable strategy to tailor the catalytic properties of molten metals and encourages further exploration of liquid metal catalysts for various other reactions.”.

Supplementary Table 1 was the most noteworthy result. In it, the authors demonstrated that Sn, SnCu, and SnIn were inactive (below detection limit); however, SnInCu was active for hexadecane dehydrogenation. It should be noted that the detection limit should be given to fully support this! Could the table or parts of it be moved to the main text?

Re: Thanks for your valuable comments. The detection limit of H₂ for the employed GC instrument is ~10 ppm, which has been updated in Table R1. Indeed, the results presented in Supplementary Table 1 are very important for this work, and some of the results have already been graphed and presented in the main manuscript. According to the reviewer’s suggestion, the results of the control experiments were included into the main manuscript (Table R1).

Table R1 | H₂ synthesis from hexadecane using different catalysts

Catalysts ^a	Reaction temperature	Hydrocarbons ^b	Reaction time	H ₂ (mol/min)	Ratio of H ₂
Sn	260 °C	hexadecane	60 min	B.D.L. ^c	n/a
SnCu _{0.0094}	260 °C	hexadecane	60 min	B.D.L.	n/a
SnIn _{0.1034}	260 °C	hexadecane	60 min	B.D.L.	n/a
Oxidized SnIn _{0.1034} Cu _{0.0094}	260 °C	hexadecane	60 min	B.D.L.	n/a
SnIn _{0.1034} Cu _{0.0094}	260 °C	hexadecane	60 min	3.6×10 ⁻⁸	~98.0%

^aReactions were performed using 5 g of bulk materials as the catalysts. ^bThe volume of hydrocarbon used for reaction was approximately 15 ml. ^cB.D.L represents beyond detection limit. The detection limit of H₂ for the employed GC instrument is ~10 ppm.

Table R1 was included in the main manuscript as Table 1 on page 8. The detection limit of H₂ was added in Table R1 as “The detection limit of H₂ for the employed GC instrument is ~10 ppm.”

Fig 3 b was also among the most noteworthy results and supports Table 1. However, other work on molten metals suggests that adsorbates and transition states can fluctuate by as much as 1 eV over time, for a given composition. So, a single configuration is less telling and other configurations should be considered, especially in light of the fact that this is one of the most noteworthy results.

Re: Thanks for your suggestions, and we agree that investigating multiple atomic configurations can provide more information, further strengthening the description of the mechanisms. We examined multiple configurations dynamically in a procedure similar to that outlined by Ruffman *et al.* (<https://doi.org/10.1039/D3SC04416E>). The obtained results suggest that a surface-exposed Cu atom and nearby In atom are key to an energetically favorable pathway, but within those criteria a wide variety of configurations remains possible. As the number of possible configurations of liquid alkane on a liquid metal surface are virtually limitless, we showed an exemplar configuration that does provide an energetically favorable pathway (Fig. 3 in the manuscript).

In order to further support the validation of proposed atomic structure and address the comment, several possible atomic configurations at the interface were investigated for H₂ synthesis from hexadecane, including Cu activating Sn and/or In rather than interacting directly. In these configurations, when an H atom was removed from C-2 in hexadecane and adsorbed on the surface, it either spontaneously returned to the C-2 during AIMD simulations or resulted in configurations that were energetically unfavourable by ~2 eV (Fig. R1). These investigations of other atomic configurations justified the validation of the proposed atomic structure for H₂ synthesis.

Fig. R1 | **a**, Exemplar configuration where manual transfer of H from C-2 resulted in the spontaneous return of H to hexadecane in geometry optimization and MD simulation. **b**, Exemplar configuration where H remains on the surface, but the relative energy compared to the reactants is +1.92 eV.

Fig. R1 was added into the Supplementary Information as Supplementary Fig. 8, and the associated discussions, including “Meanwhile, several possible atomic configurations at the interface were investigated for H₂ synthesis from hexadecane, including Cu activating Sn and/or In rather than interacting directly (Supplementary Fig. 8). In these configurations, when an H atom was removed from C-2 in hexadecane and adsorbed on the surface, it either spontaneously returned to the C-2 during AIMD simulations or resulted in configurations that were energetically unfavourable by ~2 eV. The catalytic process becomes energy favorable only when Cu is exposed on the interfacial layer and directly interacts with hexadecane.” were further included in the main manuscript on page 11.

The performance was quantified for their reactor, although the rate is not particularly high – although some comparison with other catalysts would be appreciated. The rate was given in units of mol/min for a give catalyst mass; although for their unique reactor, it is unclear if these units are best for comparison to other catalysts. Also, the conversion over the experiments should be specified. A back of the envelope calculation suggests less than 0.1%; however, this should be reported.

Re: Thanks for your suggestions, and we agree that the merits of our reaction system can be highlighted through the comparison with other catalysts. So far, both solid-state catalysts and Ga-based liquid alloys have been used for dehydrogenation reactions of hydrocarbons. As shown in Table R2, we compared the reaction conditions and efficiency of our reaction system with other catalysts for dehydrogenation reactions. Additionally, the costs of catalytic materials were discussed.

For solid-state catalysts, precious metals, such as Pt, Pd and Ru, are often involved for the preparation of catalysts. Also, high reaction temperatures, normally over 500 °C, are required to trigger the dehydrogenation reactions. Indeed, some of these catalysts demonstrated high reactivity for hydrocarbon dehydrogenations. However, the costs of these solid catalysts normally range from ~US\$25 to ~US\$100 per gram (reference: stock price Oct. 2024). The harsh operating conditions and the high cost of the catalytic materials restrict the applications of these catalysts.

A series of Ga-based liquid metal catalysts were also developed for dehydrogenation reactions (Table R2). Owing to the relatively high price of Ga, preparing one gram of these Ga-based catalysts require ~US\$4 per gram (reference: stock price Oct. 2024). High reaction temperatures are still demanded in these cases. Hence, these Ga-based alloys still face limitations for dehydrogenation reactions.

In this work, molten Sn was used as the solvent, without involving precious metals for catalyst synthesis. Synthesizing one gram of $\text{SnIn}_{0.1034}\text{Cu}_{0.0094}$ catalyst only costs ~US\$0.7 (reference: stock price Oct. 2024), which is more than >35 times cheaper than the solid-state catalysts and >5 times cheaper than Ga-based alloys presented in Table R2. Meanwhile, this Sn-based

reaction system can facilitate the dehydrogenation reactions at a lower reaction temperature of 260 °C.

The efficiency of SnIn_{0.1034}Cu_{0.0094} alloy was further calculated based on the weight of catalytic materials for the direct comparison with other materials. In our case, only the liquid atoms on the surface were involved in the catalytic reaction. Given the surface area of the SnIn_{0.1034}Cu_{0.0094} particles and the amount of produced H₂ (presented in Supplementary discussions), the efficiency of this reaction system by using canola oil as feedstock was estimated to be $\sim 9.0 \times 10^{-3} \text{ mol} \cdot \text{min}^{-1} \cdot \text{g}_{\text{catalyst}}^{-1}$ (Table R2). The efficiency of this Sn-based alloy was comparable to many of the previously reported catalysts. Despite some Pt-based solid catalysts revealed higher efficiency, the high costs of such catalytic materials and harsh operating conditions undermine many of the practical values of these catalysts. Additionally, owing to the complexity of canola oil, it is not practical to calculate the overall conversion in our reaction system.

In comparison, our approach exhibits a few obvious advantages including use of renewable feedstocks, low operating temperature and low cost of catalysts.

Table R2 was included in the Supplementary Information as Supplementary Fig. 10, and the associated discussions were presented as Supplementary discussion in the Supplementary Information. Also, the sentence “Additionally, comparing with other reported materials, molten SnIn_{0.1034}Cu_{0.0094} alloy exhibits several advantages for H₂ generation from hydrocarbons, including low cost of catalyst preparations (\sim US\$0.7/g_{catalyst}) (reference: stock price Oct. 2024), mild operating conditions and the use of renewable feedstocks (Supplementary Table 10, detailed discussions presented in Supplementary discussions).” was added into the main manuscript on page 14.

Table R2 | Comparison with reported technologies for dehydrogenation reactions

Catalytic materials	Feedstock	Reaction temperature (°C)	Pressure	Reaction types	Selectivity (%)	Efficiency (mol·min ⁻¹ ·g _{catalyst} ⁻¹) ⁽¹⁾	Ref.
Pt–Sn/CeO ₂	Propane	680	n/a	Dehydrogenation reaction	84.5	~8.0×10 ⁻²	1
[PtZn ₄]	Propane	520-620	n/a	Dehydrogenation reaction	< 95.0	~1.3	2
PtGa-Pb	Propane	600	n/a	Dehydrogenation reaction	99.6	~8.0×10 ⁻³	3
Pt-Sn/Mg-Al	Methylcyclohexane	300	n/a	Dehydrogenation reaction	~90.5	~5.0×10 ⁻³	4
Pt/TiO ₂	Methylcyclohexane	400	n/a	Dehydrogenation reaction	99.9	~2.0×10 ⁻²	5
Ga ₅₂ Pt/SiO ₂	Methylcyclohexane	450	1 bar	Dehydrogenation reaction	85.0	~7.2×10 ⁻⁵	6
Ga-Pd	Butane	445-500	1.1 bar	Dehydrogenation reaction	85.0	n/a	7
Ga–Rh	Propane	550	n/a	Dehydrogenation reaction	92.0	~5.8×10 ⁻³	8
Ga ₈₄ Pt/Al ₂ O ₃	n-heptane	410-470	n/a	Dehydrogenation reaction	~80.0	~4.5×10 ⁻⁴	9
SnIn _{0.1034} Cu _{0.0094}	Canola oil	260	n/a	Dehydrogenation reaction	~93.0	~9.0×10 ⁻³ (2)	This work

Notes: (1) The efficiencies of the catalysts were calculated based on the weight of functional materials without including the weight of supporting species. (2) Only the liquid atoms on the surface of molten alloy were included in the calculations.

The manuscript also mentions the scalability of the catalytic process but does not provide detailed quantitative analysis or comparison with other systems. This lack of detailed scalability metrics (like catalyst loading, reaction volume scaling, and economic feasibility at larger scales) may leave the reader questioning the practical applicability of the findings.

Re: Thanks for your comments. The scalability of this reaction system was initially investigated by loading 0.4, 0.6, and 0.8 g of $\text{SnIn}_{0.1034}\text{Cu}_{0.0094}$ particles on the glass microfiber filter papers. To further demonstrate the scalability of this catalytic system, we conducted the experiment on a larger scale. The experiment was performed by using 5.0 g of $\text{SnIn}_{0.1034}\text{Cu}_{0.0094}$ particles as catalytic materials in the reactor of 200 ml size. During the reaction, an H_2 production rate of 1.2×10^{-4} mol/min was obtained. A linear increase in H_2 generation was observed with reference to the amount of loading material, demonstrating the promising scalability of this reaction system (Fig. R2 and Table R3). The economic feasibility of this approach is presented in Table R2 by comparing with other solid and liquid state catalysts.

Fig. R2 | Scaled-up experiments by loading different weights of catalytic materials on glass microfiber filter papers in the reaction system.

Table R3 | Scaled-up experiments for H₂ production

Materials	Loading amount (g)	Reaction temperature	Hydrocarbon source	Volume of oil	Reaction time (h)	H ₂ (mol/min)	Ratio of H ₂
SnIn _{0.1034} Cu _{0.0094} particles on glass microfiber filter papers	5.0	260 °C	Canola oil	200 ml	1	1.2×10 ⁻⁴	~93.0%

Fig. R2 was included as Fig. 4d in the main manuscript, and Table R3 was added into Supplementary Table 8. The sentences in the abstract “Experiments with a natural oil as feedstock underscore this approach’s performance, producing 2.3×10^{-5} mol/min of H₂ with 0.8 g of catalyst at ~93.0% selectivity, while offering excellent scalability and durability at 260 °C.” was revised into “Experiments with a natural oil as feedstock underscore this approach’s performance, producing 1.2×10^{-4} mol/min of H₂ with 5.0 g of catalyst at ~93.0% selectivity, while offering reliable scalability and durability at 260 °C.” Meanwhile, the sentence “At 260 °C, H₂ production with an efficiency of 2.3×10^{-5} mol/min and a selectivity of ~93.0% was obtained using 0.8 g of SnIn_{0.1034}Cu_{0.0094} particles as catalysts.” in the main manuscript on page 4 was modified to “At 260 °C, H₂ production with an efficiency of 1.2×10^{-4} mol/min and a selectivity of ~93.0% was obtained using 5.0 g of SnIn_{0.1034}Cu_{0.0094} particles as catalysts.” Additionally, the sentence “Moreover, the setup can be readily scaled-up by loading more catalysts, with a linear increase in H₂ generation observed when using 0.4, 0.6, and 0.8 g of SnIn_{0.1034}Cu_{0.0094} particles as the catalyst (Fig. 4d, Supplementary Table 4 and Supplementary Video 1).” on page 12 was further revised into “This setup can be readily scaled-up by loading more catalysts, with a linear increase in H₂ generation observed when using 0.4, 0.6, 0.8 and 5.0 g of SnIn_{0.1034}Cu_{0.0094} particles as the catalyst (Fig. 4d, Supplementary Table 8 and Supplementary Video 1).”

The manuscript discusses AIMD simulations that probe the role of Cu and its “fluidity”. It seems that more could be done to understand why In is so important, based on the results from Table 1. For example, in line Lines 160-162: the explanation uses models which is Sn-In, Sn-Cu, and Sn only. To probe the role of In, the concentration of In should be changed to determine how that impacts the fluidity of Cu. They say In facilitates Cu’s presence at the surface as Cu diffuses away in <2 ps after In is transformed to Sn. What about the concentration dependence of In? It seems like more could be done to explain the synergy of In and Cu in the Sn solvent via AIMD simulations similar to what has already been done. For example: change only some of the In atoms to Sn and see how the system changes to investigate different In concentrations.

Re: Thank you very much for your constructive suggestions. We agree that changing only some of the In atoms to Sn could help to better understand the mechanisms and the role of In atoms during the catalytic process. Unfortunately, there are limitations with AIMD simulations, especially for the dynamicity and mobility of liquid metals. The simulation boxes are small and periodic, and the timescales are short. Within these constraints, it is difficult to see clear differences between small changes in concentration. In future work, we will examine larger systems for longer time and other liquid metal mixtures for this purpose. This will be an ongoing multi-year project and outside the scope of the current work.

A few specific minor comments:

- Fig 1c: why is Cu not included at all positions? (if 0, include as 0)

Re: Thank you for your advice. Fig. 1c has been modified as you suggested (shown as Fig. R3) and further updated in the main manuscript.

Fig. R3 | Schematics and computational modelling of molten $\text{SnIn}_{0.1034}\text{Cu}_{0.0094}$. **a**, Schematics illustrating the preparation of $\text{SnIn}_{0.1034}\text{Cu}_{0.0094}$ and the fluidity of dissolved atoms. **b**, Pairwise probability function for Sn, In, and Cu atoms at the interface of the molten alloy. **c**, Atomic density profiles for all atoms, Sn, In, and Cu as a function of z position at the interfacial layer after 200 ps. **d**, Distance between the two Cu atoms as a function of time. **e**, Distance from the interface for the two Cu atoms, where the interface is defined as the midpoint of the interfacial layer based on the atomic density profile.

• Figure 2a relates more to the group's other work than the present manuscript.

Re: Thank you for your comment. In our new work, we aimed to explore alternative metallic solvents to Ga with lower cost and distinctive catalytic properties. In our new work, we used Sn as the solvent. Owing to the inherent physiochemical properties, liquid atoms in Sn and Ga solvents demonstrate different atomic configurations, thus leading to distinct catalytic phenomena. Hydrocarbon feedstocks can be converted into propylene and H_2 , respectively, on the surface of Ga- and Sn-based liquid alloy. We believe that this mechanism and the associated catalytic phenomena can be better illustrated by presenting the different atomic structures in Ga and Sn solvents.

- Line 66-67: justify this statement with the specific observation that they authors have made to make the statement

Re: Thank you for your advice. This specific observation and the associated statements are made based on computational simulations. To justify this statement, the sentence “**In the presence of hexadecane, a Cu atom migrates and becomes transiently exposed at the interfacial layer, forming a unique structure with neighboring In and Sn atoms.**” was revised into “**According to computational simulations, a Cu atom migrates and becomes transiently exposed at the interfacial layer, forming a unique structure with neighboring In and Sn atoms in the presence of hexadecane.**”.

- Lines 62-94 has some repetitions

Re: Thank you for your comment. The repetitions were removed from the text.

- Lines 116-118: Supplementary Fig. 3 does not probe carbon via EDX, only Sn, In, Cu.

Re: Thank you for the reminder. The EDX notation of carbon on the surface of catalyst was updated in the figure (Fig. R4), which was included in the Supplementary Information as Supplementary Fig. 7.

Fig. R4 | SEM and the associated EDS images of $\text{SnIn}_{0.1034}\text{Cu}_{0.0094}$ after the reaction with hexadecane as the feedstock. After the reaction, the alloy was gently washed several times with acetone to remove the residual liquid

feedstock and then dried under vacuum. No solid byproducts were observed on the surface of the catalyst after the reaction.

- The solubility limits of Cu in the melts should be given somewhere

Re: Thank you for your suggestions. Based on the Sn-Cu phase diagram (*J. Alloys Compd.* **695**, 3666-3673, 2017), ~1 wt% of Cu can be dissolved into Sn solvent as a eutectic alloy at 260 °C. In our case, 0.5 wt% of Cu was used for preparing this Sn-based catalyst, indicating that Cu can be completely dissolved at 260 °C. The sentence “**The solubility limit of Cu in molten Sn solvent is ~1 wt% at 260 °C.**” was added into the main manuscript on page 18.

Reviewer 3:

In this manuscript, the authors used molten Sn as a metallic solvent to dissolve multiple elements, thereby lowering the melting points of the resulting alloys. It was demonstrated that the alloy $\text{SnIn}_{0.1034}\text{Cu}_{0.0094}$ selectively synthesizes hydrogen (H_2) at the interface when reacting with various hydrocarbons, including hexadecane and canola oil, at 260°C . Furthermore, an ab initio molecular dynamics (AIMD) simulation study revealed that Cu atoms migrate and transiently expose themselves at the interfacial layer. Following the adsorption of hexadecane, a Cu atom accepts a hydrogen atom, with a neighboring Sn atom stabilizing the carbon site. Concurrently, fluidic In atoms play a crucial role in ensuring Cu exposure, enhancing H_2 production in Sn-based alloys as the In concentration increases. Similarly, hydrogen production from canola oil was investigated, revealing identical reaction pathways as those observed with hexadecane.

The paper proposes expanding the use of liquid metal catalysts by exploring Sn-based alloys as a cost-effective alternative to Ga-based alloys for practical applications. However, it currently lacks sufficient experimental details and descriptions for the observed catalytic trends and does not provide adequate data to support the proposed catalytic mechanisms. This affects the reliability of the findings. Additional experimental and analytical data, including those specified below, should be provided to strengthen the paper before it can be considered for publication in Nature Communications.

Re: Thank you for your comments and suggestions. We believe that this manuscript was further strengthened after addressing your comments, and the responses to your comments are presented point-by-point.

1. In this manuscript, $\text{SnIn}_{0.1034}\text{Cu}_{0.0094}$ catalyst appears to require more harsh conditions for the H_2 synthesis compared to $\text{GaSn}_{0.029}\text{Ni}_{0.023}$. Including data on the variation in H_2 production of $\text{SnIn}_{0.1034}\text{Cu}_{0.0094}$ at different reaction temperatures would be advantageous. Does catalytic performance of $\text{SnIn}_{0.1034}\text{Cu}_{0.0094}$ decrease dramatically at temperatures lower than 260°C ?

Re: Thank you for your comments, and you are right about the reaction temperatures. Compared to Ga solvent, a high temperature is required to maintain the molten state of $\text{SnIn}_{0.1034}\text{Cu}_{0.0094}$ alloy owing to its higher melting point. At 260°C , the specific structure of freely moving atoms, in liquid state, on the surface of molten $\text{SnIn}_{0.1034}\text{Cu}_{0.0094}$ alloy ensures a unique adsorption pattern, when hexadecane is in the vicinity of the surface, and energy-favourable reaction pathways for H_2 production.

For addressing your comment, we further explored the influence of reaction temperatures and conducted the experiments at 150 and 200°C , respectively. Based on our observations,

SnIn_{0.1034}Cu_{0.0094} alloy is in solid state at both 150 and 200 °C. With hexadecane as feedstock, no obvious H₂ production was observed at the reaction temperature of 150 and 200 °C (Table R4). These results indicate that liquid state of SnIn_{0.1034}Cu_{0.0094} alloy is the prerequisite for H₂ generation.

Table R4 | The influence of reaction temperature for H₂ synthesis

	Materials	Reaction temperature	Hydrocarbon source	Volume	Reaction time	H ₂ (mol/min)	Ratio of H ₂
Bulk materials	SnIn _{0.1034} Cu _{0.0094} 5g	150 °C	hexadecane	15 ml	60 min	B.D.L. ⁽¹⁾	n/a
	SnIn _{0.1034} Cu _{0.0094} 5g	200 °C	hexadecane	15 ml	60 min	B.D.L.	n/a
	SnIn _{0.1034} Cu _{0.0094}	260 °C	hexadecane	60 min	60 min	3.6×10 ⁻⁸	~98.0%

Note:

(1) B.D.L. represents beyond detection limits.

Table R4 was included in the Supplementary Information as Supplementary Table 2. The sentences “Additionally, no obvious H₂ generation was detected by conducting the experiments at the reaction temperature of 150 and 200 °C, which are below the melting point of SnIn_{0.1034}Cu_{0.0094} alloy (Supplementary Table 2). This observation suggests that liquid state of SnIn_{0.1034}Cu_{0.0094} alloy is the prerequisite for H₂ generation.” were added into the manuscript on page 8.

2. In Figure 2d and 4b, the systems with higher In content, specifically SnIn_{0.1034}Cu_{0.0094} and SnIn_{0.5169}Cu_{0.0094}, demonstrate both high selectivity and significant H₂ production. The authors should provide a detailed explanation from the perspective of “catalytic selectivity”. This should include an analysis of how Cu and In contribute to the observed catalytic behavior, particularly regarding the selective synthesis of H₂ and the overall catalytic efficiency.

Re: Thank you for your comments. Within molten SnIn_{0.1034}Cu_{0.0094} alloy, Cu atoms can migrate and reach to the interface in the presence of In atoms. During the reaction process, the role of Cu atom is to accept an H from the reactant, and the role of In is favoured for both stabilizing intermediate and facilitating the presence of Cu at the interface. The synergy

between In and Cu atoms ensures an energy favorable and selective reaction pathway for H₂ production. Without the presence of In atoms, Cu diffused away from the interface in less than 2 ps in our simulations. According to our control experiments and associated computational simulations presented in Fig. 3, the synergy between Cu and In is essential for the reaction.

As your request, the discussions regarding the synergistic role of In and Cu and the energy barriers for H₂ production, including “As shown in Fig. 3b, the reaction profiles were recalculated with Cu atoms alchemically transformed to Sn (*i.e.*, Sn and In only), with In atoms alchemically transformed to Sn (*i.e.*, Sn and Cu atoms only), and with both In and Cu atoms alchemically transformed to Sn (*i.e.*, Sn atoms only). In the Sn and In only system, the removal of H from C-2 changes the process from essentially energetically neutral in the presence of Cu to a process that is energetically unfavorable by 0.56 eV. While the product configuration is 0.34 eV lower in energy than the intermediate, it is still 0.22 eV higher in energy than the reactants. This is in line with the experimental results. In the absence of In (*i.e.*, Sn-Cu and Sn only systems), the removal of H from C-2 rises to 0.86 eV for Sn-Cu and 1.62 eV for Sn only, in line with the experimental observations. These results highlight the synergistic role of In and Cu in stabilizing H on the surface, and the dynamic role of Cu in facilitating an energetically favorable process overall. To further investigate the role of In, snapshots from AIMD simulations were taken where the Cu was exposed at the interface in the presence of hexadecane, alchemically transformed In to Sn, and continued the simulations. In all cases, the Cu diffused away from the interface in less than 2 ps, indicating that the role of In may be both stabilizing H on the surface and facilitating the presence of Cu at the interface.” were included in the main manuscript on page 10.

Additionally, we further investigated several other configurations between Sn, In and Cu atoms, including Cu activating Sn and/or In rather than interacting directly. In these configurations, when an H atom was removed from C-2 in hexadecane and adsorbed on the surface, it either

spontaneously returned to the C-2 during AIMD simulations or resulted in configurations that were energetically unfavourable by ~2 eV (Fig. R5).

Fig. R5 | **a**, Exemplar configuration where manual transfer of H from C-2 resulted in the spontaneous return of H to hexadecane in geometry optimization and MD simulation. **b**, Exemplar configuration where H remains on the surface, but the relative energy compared to the reactants is +1.92 eV.

Fig. R5 was added to the Supplementary Information as Supplementary Fig. 8. Also, the associated discussions of Fig. R5, including “Meanwhile, several possible atomic configurations at the interface were investigated for H₂ synthesis from hexadecane, including Cu activating Sn and/or In rather than interacting directly (Supplementary Fig. 8). In these configurations, when an H atom was removed from C-2 in hexadecane and adsorbed on the surface, it either spontaneously returned to the C-2 during AIMD simulations or resulted in configurations that were energetically unfavourable by ~2 eV. The catalytic process becomes energy favorable only when Cu is exposed on the interfacial layer and directly interacts with hexadecane.” were further included in the main manuscript on page 11.

3.The author prepared the Sn-based alloys with varying In ratios, including SnIn_xCu_{0.0094} (x=0.0103, 0.0518, 0.1034). In AIMD simulations, the proposed catalytic mechanisms suggests

that the Cu and Sn serve as the active catalytic sites, with Cu adsorbing hydrogen (H) and Sn adsorbing carbon (C). Therefore, controlling the amount of Cu in the alloys could provide strong evidence supporting the observed catalytic tendencies and mechanisms.

Re: Thank you for your suggestions. We agree that the proposed catalytic mechanisms can be further strengthened by analysing different concentrations of dissolved Cu in the molten alloys. Hence, a series of alloys containing different amount of Cu, including $\text{SnIn}_{0.1034}\text{Cu}_{0.0047}$, $\text{SnIn}_{0.1034}\text{Cu}_{0.0018}$, and $\text{SnIn}_{0.1034}\text{Cu}_{0.0009}$, were synthesized (Table R5). With hexadecane as hydrocarbon feedstock, the reactivity and selectivity conspicuously decreased when $\text{SnIn}_{0.1034}\text{Cu}_{0.0047}$ was used as the catalyst at 260 °C. Hydrogen production was not achieved by using $\text{SnIn}_{0.1034}\text{Cu}_{0.0018}$, and $\text{SnIn}_{0.1034}\text{Cu}_{0.0009}$ as the catalysts, respectively. The decreased reactivity possibly originates from the lower accessibility of Cu atoms on the surface of molten alloys. These results further supported the proposed mechanisms regarding the synergy between Cu and In atoms.

Table R5 | H₂ synthesis using catalysts containing different amount of Cu

	Materials	Reaction temperature	Hydrocarbon source	Volume	Reaction time	H ₂ (mol/min)	Ratio of H ₂
Bulk materials (5 g)	$\text{SnIn}_{0.1034}\text{Cu}_{0.0009}$	260 °C	hexadecane	15 ml	60 min	B.D.L. ⁽¹⁾	n/a
	$\text{SnIn}_{0.1034}\text{Cu}_{0.0018}$	260 °C	hexadecane	15 ml	60 min	B.D.L.	n/a
	$\text{SnIn}_{0.1034}\text{Cu}_{0.0047}$	260 °C	hexadecane	15 ml	60 min	7.2×10^{-9}	~85.0%
	$\text{SnIn}_{0.1034}\text{Cu}_{0.0094}$	260 °C	hexadecane	15 ml	60 min	3.6×10^{-8}	~98.0%

Note:

(1) B.D.L. represents beyond detection limits.

Table R5 was presented in Supplementary Information as Supplementary Table 3. The sentences “To confirm the role of Cu atoms in the reaction, a series of alloys with varying Cu concentrations, including $\text{SnIn}_{0.1034}\text{Cu}_{0.0047}$, $\text{SnIn}_{0.1034}\text{Cu}_{0.0018}$ and $\text{SnIn}_{0.1034}\text{Cu}_{0.0009}$, were synthesized. The reactivity and selectivity of these molten alloys for H₂ generation reduced as

Cu concentration decreased, with $\text{SnIn}_{0.1034}\text{Cu}_{0.0018}$ and $\text{SnIn}_{0.1034}\text{Cu}_{0.0009}$ failing to produce H_2 from hexadecane (Supplementary Table 3). The lowered efficiency is likely due to the lesser availability of Cu atoms on the surface of the molten alloys. These findings indicate that the presence of surface Cu atoms and the interaction between In and Cu are crucial for the reaction.”

were added into the main manuscript on page 9.

4. The catalyst droplets with an average diameter of 1200 nm were used in the H_2 production tests. If the size of these droplets can be controlled, the effect of droplet size on catalytic performance should be described. Additionally, providing size distribution data of the droplets after the catalytic measurements is necessary to confirm whether aggregation occurred during the catalytic reactions. This information is crucial for understanding the impact of droplet size on catalytic efficiency and stability, as well as for optimizing the catalyst design.

Re: Thank you for your constructive suggestions. The particles of molten $\text{SnIn}_{0.0207}\text{Cu}_{0.0094}$ were loaded on the glass microfiber filter papers to avoid aggregation during the reaction process and increase the surface area for reaction. We further measured the size of $\text{SnIn}_{0.1034}\text{Cu}_{0.0094}$ particles after the reaction. No obvious size change was observed after the reaction (medium size of ~1200 nm), indicating that the catalytic materials aggregation did not occur during the reaction (Fig. R6).

To further explore the effects of droplet size, we synthesized $\text{SnIn}_{0.1034}\text{Cu}_{0.0094}$ particles of different sizes for H_2 generation from hydrocarbons. The size of $\text{SnIn}_{0.1034}\text{Cu}_{0.0094}$ particles can be to some degree regulated by controlling the sonication time. After sonication for 10 and 20 min, the average particle diameters were measured to be ~1850 and ~1600 nm, respectively (Table R6). By using canola oil as hydrocarbon feedstock, H_2 production was observed by using $\text{SnIn}_{0.1034}\text{Cu}_{0.0094}$ particles of different droplet sizes as catalysts at 260 °C. It is seen that the exposed surface area of $\text{SnIn}_{0.1034}\text{Cu}_{0.0094}$ increases as the average particle size decreases. As illustrated in Fig. R7, a reduction in particle size resulted in increased H_2 generation rates. These findings suggest that the exposed surface area of the catalysts is a key factor in determining the reaction system's efficiency.

Fig. R6 | Size distributions of SnIn_{0.1034}Cu_{0.0094} particles after reaction (sonication for 30 min initially).

Fig. R7 | H₂ production by using SnIn_{0.1034}Cu_{0.0094} particles of different sizes. The particles were prepared by using a probe sonicator (model VC 750 from Sonics & Materials) under the protection of N₂ while being heated to ~300 °C to keep the alloy molten. The sonication ultrasonic power input was set to ~410 W, with a 1 s pause after every 9 s of sonication. The sonication time was set for 10, 20 and 30 min to obtain SnIn_{0.1034}Cu_{0.0094} particles of different sizes. As the particle size decreases, the exposed surface area of SnIn_{0.1034}Cu_{0.0094} increases.

Table R6 | H₂ synthesis using catalysts of different particle sizes

	Sonication time (min)	Medium diameter (nm) ⁽¹⁾	Reaction temperature	Hydrocarbon source	H ₂ (mol/min)	Ratio of H ₂
SnIn _{0.1034} Cu _{0.0094} particles on glass microfiber filter papers (0.2 g)	30	1200	260 °C	Canola oil	6.0×10 ⁻⁶	~93.0%
	20	1600	260 °C	Canola oil	5.4×10 ⁻⁶	~93.0%
	10	1850	260 °C	Canola oil	4.2×10 ⁻⁶	~93.0%

Note: (1) The particles were prepared by using a probe sonicator (model VC 750 from Sonics & Materials) under the protection of N₂ while being heated to ~300 °C to keep the alloy molten. The sonication ultrasonic power input was set to ~410 W, with a 1 s pause after every 9 s of sonication. The sonication time was set for 10, 20 and 30 min to obtain SnIn_{0.1034}Cu_{0.0094} particles of different sizes.

Fig. R6 and R7 were included in the Supplementary Information as Supplementary Fig. 11 and 12, respectively. Table R6 was added into the Supplementary Information as Supplementary Table 7. The sentence “Meanwhile, there was no noticeable change in the size of the SnIn_{0.1034}Cu_{0.0094} particles after the reaction, suggesting that the molten droplets remained separate and did not aggregate during the process (Supplementary Fig. 11). For comparison, the SnIn_{0.1034}Cu_{0.0094} particles of different diameters were also synthesized (Supplementary Table 7). The exposed surface area of SnIn_{0.1034}Cu_{0.0094} increased as the particle size decreased, and a reduction in particle size resulted in increased H₂ generation rates (Supplementary Fig. 12). These results reveal that the exposed surface area of the catalysts is a key factor in determining the reaction system's H₂ production rate.” were presented in the main manuscript on page 12.

5. The manuscript states that In is found in regions with relatively low tin, Sn, density, while Cu is primarily located near the bottom of the interfacial layer, occasionally becoming transiently exposed at the interface in the presence of hexadecane. However, the SEM images in the supplementary information do not adequately support this explanation. Additional data, such as high-resolution imaging or compositional analysis, is needed to substantiate these

claims and provide a clearer understanding of the spatial distribution and interaction of In and Cu within the catalyst.

Re: Thank you for your constructive comments, and we agree with your points. Originating from their dynamicity, mechanistic investigations regarding the atomic configurations and the catalytic behavior of liquid atoms are challenging. Current interpretations, such as the growth of diamond (*Nature*, 629, 348–354, **2024**), methane pyrolysis (*Science*, 381, 857-861, **2023**) and ammonia synthesis (*Nat. Catal.*, 7, 1044–1052, **2024**), are mainly based on computational simulations.

To address your comment and further validate the proposed atomic arrangements on the surface of liquid metals, we tried to perform the scanning tunnelling microscopy (STM). However, owing to the instability of the liquid metal surface at the atomic level at high temperature and the restrictions of the STM facility, visualizing the atomic configurations at the interfaces was inconclusive.

As an alternative surface characterisation method, cyclic voltammetry was conducted to investigate atomic distributions on the surface of molten alloys. Molten Sn, SnCu_{0.0094}, SnIn_{0.1034} and SnIn_{0.1034}Cu_{0.0094} were prepared and rapidly frozen to serve as the working electrodes. By using Sn as the working electrode, the oxidation peak was observed at around -0.85 V (Fig. R8a). Using SnCu_{0.0094} as the working electrode, the peak at -0.85 V was assigned to the oxidation of Sn (Fig. R8b). No additional significant oxidation peak was observed, suggesting that Cu atoms remain beneath the interfacial layer within the Sn solvent. The peaks centered at -0.85 V and 1.10 V can be assigned to the oxidation of Sn and In, respectively, when SnIn_{0.1034} was employed as the working electrode (Fig. R8c). This observation indicates that In atoms can remain on the interfacial layer within molten Sn. Notably, one additional peak at 0.21 V was observed by using SnIn_{0.1034}Cu_{0.0094} as the working electrode, originating from the oxidation of Cu (Fig. R8d). These results suggested that Cu atoms can reach to the interfacial layer in the presence of In atoms on the surface of molten SnIn_{0.1034}Cu_{0.0094} alloy. The cyclic

voltammetry experiment outcomes regarding the relative distributions of liquid In and Cu atoms in molten Sn solvent are in accordance with the computational simulations, further confirming our proposed reaction mechanisms.

Fig. R8 | Cyclic voltammograms by employing different alloys as the working electrode. a-d, Cyclic voltammograms obtained by using Sn (a), SnCu_{0.0094} (b), SnIn_{0.1034} (c) and SnIn_{0.1034}Cu_{0.0094} (d) as the working electrodes. The cyclic voltammetry experiments were conducted to investigate the relative positions of the dispersed atoms in molten SnIn_{0.1034}Cu_{0.0094} alloy with reference to the surface and in comparison to Sn, SnCu_{0.0094}, and SnIn_{0.1034}. Here, Sn, SnCu_{0.0094}, SnIn_{0.1034} and SnIn_{0.1034}Cu_{0.0094} were painted onto an indium tin oxide (ITO) substrate at 260 °C. After quick freeze of the liquid alloys using liquid nitrogen, the solidified Sn, SnCu_{0.0094}, SnIn_{0.1034} and SnIn_{0.1034}Cu_{0.0094} on the substrates were used as the working electrodes. The experiments were conducted under the same conditions by using a calomel reference electrode and a gold counter electrode to set up a three-electrode configuration. An acetonitrile solution containing 0.10 M tetrabutylammonium hexafluorophosphate was used as the electrolyte.

Fig. R8 was included in the Supplementary Information as Supplementary Fig. 1, and the associated discussions were presented in Supplementary discussions in the Supplementary Information. Also, the sentences “Cyclic voltammetry was performed to explore the distributions of Cu and In atoms within molten Sn solvent and with reference to their surfaces. The results revealed that Cu atoms remained below the interfacial layer of molten Sn and could only reach to the surface in the presence of In atoms (Supplementary Fig. 1, detailed discussions presented in Supplementary discussions). These observations are in accordance with the computational simulations.” were included in the main manuscript on page 6.

6. The energy barriers for H₂ synthesis are presented in Figure 3b, illustrating the hydrogen transfer and H₂ formation energies. However, the figure lacks descriptors for the adsorption energies of the reactants. The adsorption step is crucial for determining the overall reaction kinetics, so the authors should provide some description of the adsorption rate.

Re: We thank the reviewer for the comment. Indeed, the adsorption step can be crucial for the overall reaction kinetics. Unfortunately, we often face limitations regarding AIMD simulations, especially for the dynamicity and mobility of liquid metals. The simulation boxes are small and periodic, and the timescales are short. Due to the prohibitive computational cost of modelling liquid hexadecane interfaced with liquid metal, we are only able to model a single hexadecane molecule. While we agree that the adsorption step can be critical, in exploring reaction mechanisms we focus on the possible steps following adsorption, due to the aforementioned computational limitations. We hope to explore larger systems with full liquid-liquid interfaces in future work. Additionally, a recently-published paper suggests that for some reactions on liquid metals, the reactant does not directly adsorb to the liquid metal surface (<https://doi.org/10.1002/anie.202407124>).

7. The authors noted that using canola oil as a feedstock resulted in an efficiency approximately ~10 times higher than when using hexadecane. However, the manuscript does not provide an explanation for this phenomenon. Since canola oil was mostly composed of fatty acids, the cause of this efficiency improvement may be due to fatty acids. Therefore, the authors should

be able to reveal this enhancement by using some fatty acids (e.g., oleic acid or α -linolenic acid) as feedstock in this reaction.

Re: Thank you for your suggestions. We agree that using oleic acid as feedstock, to some extent, can help us understand the enhanced efficiency for canola oil. To examine your suggestion, we conducted the experiment suggested in your comment. Using oleic acid as the hydrocarbon feedstock, a significant increase in H₂ production at the rate of 1.8×10^{-5} mol/min was observed with 0.2 g of SnIn_{0.1034}Cu_{0.0094} particles as the catalyst at 260 °C. Compared with those of using hexadecane and canola oil as feedstock, the efficiency was ~30 and ~3 times higher, respectively. Meanwhile, the selectivity for H₂ using oleic acid as feedstock was ~84.5%, which is lower than that of hexadecane and canola oil cases (Table R7).

Indeed, the rate of H₂ production by using oleic acid as feedstock was significantly enhanced, which possibly originated from its smaller molecular size and carboxyl group (-COOH). These observations can partially explain that the functional group (-COOC-) in canola oil possibly boost the generation of H₂ during the reaction. However, it is also important to note that despite their structural similarities, canola oil and oleic acid are not considered as the same species as the result of their different functional groups. Canola oil is mainly composed of triacylglycerols (-COOC-), and the molecular formula, C₅₇H₁₁₀O₆, is often used to represent canola oil for simplicity. The molecular formula of oleic acid is C₁₈H₃₄O₂ with a different functional group (-COOH).

Table R7 | The influence of hydrocarbon feedstocks for H₂ synthesis

	Materials	Reaction temperature	Hydrocarbon source	Volume	Reaction time	H ₂ (mol/min)	Ratio of H ₂
Particles loaded on glass microfiber filter papers	SnIn _{0.1034} Cu _{0.0094} 0.2g	260 °C	hexadecane	15 ml	60 min	6.0×10^{-7}	~98.0%
	SnIn _{0.1034} Cu _{0.0094} 0.2g	260 °C	oleic acid	15 ml	60 min	1.8×10^{-5}	~84.5%
	SnIn _{0.1034} Cu _{0.0094} 0.2g	260 °C	canola oil	15 ml	60 min	6.0×10^{-6}	~93.0%

Table R7 was included the Supplementary Information as Supplementary Table 9. The sentences “To investigate the reason for enhanced efficiency of H₂ generation in canola oil case, oleic acid was further selected as the feedstock owing to some structural similarity to canola oil. Under the same reaction conditions, an increased H₂ generation rate of 1.8×10^{-5} mol/min, with the selectivity of ~84.5%, was obtained by using oleic acid hydrocarbon source (Supplementary Table 9). The increased reaction rate is likely attributed to the presence of the carboxyl group in oleic acid, which partially explains the higher efficiency observed with canola oil, given their structural similarity.” were presented in the main manuscript on page 13.

Reviewer 4:

The authors have performed experimental and theoretical studies to explore the reactivity of a Sn-based ternary alloy at low temperatures. For this, they have used natural oil feedstock for producing H₂. It is not clear to me what is the novelty of this work. Several works are showing Sn as a solvent liquid for active metals for the catalysis of hydrocarbons to H₂. The authors may want to look at the works of Ogino et al. from the 1970s to 1980s or more recent works from the McFarland group at UCSB.

Re: Thank you for your comments. Indeed, molten metals, such as Ni-Bi alloy (*Science*, 358, 917-921, **2017**), ternary NiMo-Bi alloy (*Science*, 381, 857-861, **2023**), and Ni-In alloy (*Nat. Catal.*, 3, 83–89, **2020**) have been used as catalysts for CH₄ pyrolysis or dry reforming. However, these reactions are normally operated at high temperatures of ~1000 °C to show suitable performances.

Another category of Ga-based liquid metals has also been used for catalytic applications, including the growth of diamond (*Nature*, 629, 348–354, **2024**), dechlorination of polyvinyl chloride (*Sci. Adv.*, 10, eadm9963, **2024**) and formate electrosynthesis (*Adv. Funct. Mater.*, 2408966, **2024**). However, the relatively high cost of Ga restricts its practical applications.

Hence, exploring molten metal catalysts, with low costs, mild operating conditions and distinctive catalytic performance, are all of great importance. In this study, we investigated an alternative Sn solvent to Ga, which is significantly less expensive and can dissolve various elements while reducing the melting points of the resulting alloys to below 250°C. By dissolving In and Cu into Sn solvent, H₂ production with a selectivity of ~93% was achieved at the reaction temperature of 260 °C using canola oil as a feedstock. Importantly, the underpinning reaction mechanisms on the surface of molten Sn alloy, regarding the unique atomic structures and reaction pathways, had not been reported. Considering such critical novelty aspects, our work has the potential to lead to new avenues for catalysts design by tuning the metallic solvents, and encourages the further investigations of liquid metal catalysts for many other applications.

The main conclusions of their paper have been drawn from simulations. However, I have serious concerns over the methodology used in the paper. Some of them are listed below:

1. It is difficult to understand how reaction barriers are obtained. The paper cited by authors to get reaction barriers only discusses getting dispersion in adsorption energies. The surface of the liquid is ever-changing. The authors also point this out and is the main conclusion of their paper. However, they report barriers based on one snapshot. Are the fluctuations or dynamic configurations not important? As per their main conclusions, dynamicity plays a major role. I would suggest authors perform more thorough free energy calculations using rare-event methods before this publication can be accepted.

Re: Thank you for your comments. We agree that using rare-event methods can provide more details regarding activation energy and other free energy calculations. Unfortunately, originating from the dynamicity and mobility of liquid atoms, AIMD simulations face limitations for molten alloys. The simulation boxes of liquid metallic atoms are small and periodic, and the timescales are short. Moreover, due to the prohibitive computational cost of modelling liquid hexadecane interfaced with liquid metal, we are only able to model a single hexadecane molecule. Due to these limitations, rather than attempting to report barriers from a single snapshot, our aim was to provide an exemplar configuration of a viable pathway and then examine how alchemically changing Cu and In to Sn affects the barrier of this pathway, similar to our previous work (<https://www.nature.com/articles/s41565-023-01540-x>). We did investigate multiple configurations, and we found that configurations similar to the one shown in Fig. 3 gave similar energies, while other configurations we investigated either spontaneously returned to reactants in AIMD simulation or gave energies ~ 2 eV higher than reactant configurations.

Other studies that use free energy or metadynamics methods (e.g., <https://pubs.acs.org/doi/10.1021/acs.jctc.0c00486> and <https://doi.org/10.1002/anie.202407124>) examine much simpler reactions with fewer than 100 atoms. Our system has 200 liquid metal atoms and an additional 50 atoms in hexadecane. Moreover, obtaining a reaction barrier from

free energy calculations for a single conformation of hexadecane combined with a single configuration of a liquid metal interface would not necessarily describe all possible pathways. For thorough investigation of liquid metal catalysis, larger systems for longer times are required, but it will be an ongoing multi-year project and outside the scope of this current work.

In order to address your comment, and considering the above limitations, instead of attempting to determine the reaction barrier from computational methods, we determined it experimentally. The Arrhenius equation was employed in this case to gain comprehensive understandings of the activation energy. We conducted the experiments at different reaction temperatures by using SnIn_{0.1034}Cu_{0.0094} alloy as catalyst and hexadecane as feedstock. As shown in Table R8, the efficiency for H₂ production increased at higher reaction temperatures. Based on these experimental results, a plot of $\ln k$ vs. $1/T$ was obtained using the following Arrhenius equations (R1 and R2):

$$k = Ae^{\frac{-E_a}{RT}} \quad (\text{R1})$$

$$\ln k = \frac{-E_a}{RT} + \ln A \quad (\text{R2})$$

where k represents the rate constant, E_a is the activation energy, R is the gas constant (8.3145 J/K mol), T is the temperature expressed in Kelvin, and A is the Arrhenius or frequency factor. According to the Arrhenius plot presented in Fig. R9, $-\frac{E_a}{R}$ equals to -12860 K, and E_a is calculated to be $\sim 1.06 \times 10^5$ J/mol. Therefore, the activation energy for a single molecule is estimated to be around 1.10 eV.

Fig. R9 | Arrhenius plot for the calculation of activation energy. The slope of the plot is calculated to be -12860 K in this case.

To further validate the proposed reaction mechanisms, other possible atomic configurations at the interface were investigated for H₂ synthesis from hexadecane, including Cu activating Sn and/or In rather than interacting directly. In these configurations, when an H atom was removed from C-2 in hexadecane and adsorbed on the surface, it either spontaneously returned to the C-2 during AIMD simulations or resulted in configurations that were energetically unfavourable by ~2 eV (Fig. R10).

In summary, the activation energy for one molecule based on the proposed atomic configurations is calculated experimentally to be ~1.10 eV, which is lower than other possible mechanism investigated (~2 eV from simulations). These results provide more details regarding activation energy, deepening our understandings of this reaction and the proposed reaction mechanisms.

Fig. R10 | **a**, Exemplar configuration where manual transfer of H from C-2 resulted in the spontaneous return of H to hexadecane in geometry optimization and MD simulation. **b**, Exemplar configuration where H remains on the surface, but the relative energy compared to the reactants is +1.92 eV.

Table R8 | The influence of reaction temperature

Materials	Reaction temperature (K)	Hydrocarbon source	Volume	Reaction time	H ₂ (mol/s)
SnIn _{0.1034} Cu _{0.0094} (bulk 5g)	543	hexadecane	15 ml	60 min	9.4×10 ⁻¹⁰
	533	hexadecane	15 ml	60 min	6.0×10 ⁻¹⁰
	523	hexadecane	15 ml	60 min	3.8×10 ⁻¹⁰

Fig. R9 and R10 were presented as Supplementary Fig. 8 and Fig. 9, respectively, in the Supplementary Information. Table R8 was included in the Supplementary Information as Supplementary Table 4. Meanwhile, the sentences “To gain a more comprehensive understanding of this reaction, the Arrhenius equation was employed to investigate the activation energy. An Arrhenius plot was generated by performing experiments at varying reaction temperatures (Supplementary Fig. 9 and Supplementary Table 4, detailed calculations provided in the Supplementary discussions). The activation energy was calculated to be approximately 1.10 eV, which is lower than that of other simulated reaction pathways requiring

at least ~2 eV (Supplementary Fig. 8). These findings offer additional insight into the activation energy, further reinforcing the proposed reaction mechanisms.” were presented in the main manuscript on page 11. Additionally, the calculations and the associated discussions were presented as Supplementary discussions in Supplementary Information.

2. The authors have not taken dispersions in their simulations. It is well known this could lead to significant errors.

Re: Thank you for your comment. We used the DFT-D3 with Becke-Johnson damping function (IVDW=12 in VASP) dispersion correction for all simulations and geometry optimizations. We apologize for not including this in the methods section, it was simply an oversight. The DFT-D3 method with Becke-Johnson damping function was used for all calculations.

The sentence “The DFT-D3 method with Becke-Johnson damping function (IVDW=12 in VASP) dispersion correction was used for all simulations and geometry optimizations.” was presented in Method section in the main manuscript on page 18.

3. It is unclear what ensemble they are working in and how the equilibration is performed. The authors mention AIMD simulation at 260 deg C. How are the authors controlling the temperature?

Re: Thank you for your comments. We performed simulations with the Nose-Hoover thermostat. As a test, we also scaled the velocities at each step (SMASS=-1 in VASP) and found no significant difference in the results. We apologize for not including this in the methods section. The sentence “All simulations were performed in the NVT ensemble with the temperature controlled by the Nose–Hover thermostat.” was added to the Method section on page 18 for clarification.

Regarding equilibration, we mention in the method that “initial configurations of hexadecane were added to equilibrated snapshots of the liquid metal” to indicate that hexadecane was added to liquid metal systems that had already been run for 200 ps rather than systems that had not

been run in AIMD. No additional distinct equilibration step was required. Hence, the sentence “For simulations involving hexadecane, random initial configurations of hexadecane were added to equilibrated snapshots of the liquid metal interface where at least one Cu was present in the interfacial layer.” in the Method section on page 17 was revised into “For simulations involving hexadecane, random initial configurations of hexadecane were added to snapshots of the liquid metal interface following 200 ps of AIMD simulation where at least one Cu was present in the interfacial layer.” for clarification.

References:

1. Xiong H., Lin, S., Goetze, J., Pletcher, P., Guo, H., Kovarik, L., Artyushkova, K., Weckhuysen, B. M., Datye, A. K. Thermally stable and regenerable platinum–tin clusters for propane dehydrogenation prepared by atom trapping on ceria. *Angew. Chem. Int. Ed.* **56**, 8986-8991 (2017).
2. Chen S., Zhao, Z.-J., Mu, R., Chang, X., Luo, J., Purdy, S. C., Kropf, A. J., Sun, G., Pei, C., Miller, J. T., Zhou, X., Vovk, E., Yang, Y., Gong, J. Propane dehydrogenation on single-site [PtZn₄] intermetallic catalysts. *Chem.* **7**, 387-405 (2021).
3. Nakaya Y., Hirayama, J., Yamazoe, S., Shimizu, K.-i., Furukawa, S. Single-atom Pt in intermetallics as an ultrastable and selective catalyst for propane dehydrogenation. *Nat. Commun.* **11**, 2838 (2020).
4. Yan J., Wang, W., Miao, L., Wu, K., Chen, G., Huang, Y., Yang, Y. Dehydrogenation of methylcyclohexane over PtSn supported on MgAl mixed metal oxides derived from layered double hydroxides. *Int. J. Hydrog. Energy.* **43**, 9343-9352 (2018).
5. Yang X., Song, Y., Cao, T., Wang, L., Song, H., Lin, W. The double tuning effect of TiO₂ on Pt catalyzed dehydrogenation of methylcyclohexane. *Mol. Catal.* **492**, 110971 (2020).
6. Sebastian O., Nair, S., Taccardi, N., Wolf, M., Søgaard, A., Haumann, M., Wasserscheid, P. Stable and selective dehydrogenation of methylcyclohexane using supported catalytically active liquid metal solutions – Ga₅₂Pt/SiO₂ SCALMS. *ChemCatChem.* **12**, 4533-4537 (2020).
7. Taccardi N., Grabau, M., Debuschewitz, J., Distaso, M., Brandl, M., Hock, R., Maier, F., Papp, C., Erhard, J., Neiss, C., Peukert, W., Görling, A., Steinrück, H. P., Wasserscheid, P. Gallium-rich Pd–Ga phases as supported liquid metal catalysts. *Nat. Chem.* **9**, 862-867 (2017).
8. Raman N., Maisel, S., Grabau, M., Taccardi, N., Debuschewitz, J., Wolf, M., Wittkämper, H., Bauer, T., Wu, M., Haumann, M., Papp, C., Görling, A., Spiecker, E., Libuda, J., Steinrück, H.-P., Wasserscheid, P. Highly effective propane dehydrogenation using Ga–Rh supported catalytically active liquid metal solutions. *ACS Catal.* **9**, 9499-9507 (2019).
9. Sebastian O., Al-Shaibani, A., Taccardi, N., Haumann, M., Wasserscheid, P. Kinetics of dehydrogenation of n-heptane over GaPt supported catalytically active liquid metal solutions (SCALMS). *React. Chem. Eng.* **9**, 1154-1163 (2024).

Reviewer #1 (Remarks to the Author):

The authors have addressed the issues raised in the previous round of revisions. I have one final minor comment: on p.3, line 52, the authors compare molten Sn with Ga, and I believe a discussion of the surface properties of these two materials would be beneficial, as it is relevant to potential applications (such as catalysis). For instance, Ga forms a self-passivating Ga_2O_3 layer spontaneously with a negative Gibbs free energy (refs: ACS Appl. Mater. Interfaces 2018, 10, 40, 34758–34764; Small, 2020, 16, 12, 1903391). In contrast, while Sn also spontaneously forms oxides (SnO or SnO_2) with negative Gibbs free energies, the oxide layer on tin does not fully passivate the surface. This non-passivating behavior arises from the properties of the tin oxide layer, which is often porous and lacks sufficient adherence to prevent further oxidation. As a result, oxidation continues gradually, particularly under conditions of high humidity or elevated temperatures (ref: Progress in Surface Science, 2005, 79, 2-4, 47-154). Including a discussion on this distinction, with relevant citations, would add depth to the comparison. Once again, this is an excellent study, and I enjoyed reading it!

Re: We sincerely thank the reviewer for the continued support. Indeed, the differences between Sn and Ga in forming different types of oxide layers can influence the surface properties of their associated alloys. We agree that further discussion on this topic can deepen the understandings of Sn-based and Ga-based liquid metal catalysts. Hence, the discussions “**The distinct surface properties of Sn and Ga may also impact the catalytic performance of their respective liquid alloys. For instance, although oxide layers spontaneously form on both Ga and Sn surfaces with negative Gibbs free energies, the oxide layer on the Sn surface does not fully passivate the interface, potentially exhibiting enhanced surface access for catalytic activities¹⁻³.**” were included in the main manuscript on page 3.

Reviewer #2 (Remarks to the Author):

I have only one further comment related to the authors responses.

While the authors added valuable information in Table R2, it would be unfair to list 9 catalysts that have rates (“efficiencies”) per g_catalyst, but then list their SnInCu catalyst with a rate per g_surface atoms. Is that how the values are calculated? If so, I suggest revising to have an apples-to-apples comparison (i.e. per g_catalyst in all cases or change the rate units all to be per square meter). As it stands, one cannot compare SnInCu to any of the other catalysts. Also, the comparison would be significantly improved if made to other catalysts for canola oil dehydrogenation, instead of other feedstocks.

Using the values in the new Table R1 of 3.6×10^{-8} (mol H₂/min) divided by 5 grams as listed in the table legend, one gets 7×10^{-9} mol/min/g_cat, NOT the value listed in Table R2. The value from Table R2 is also different from what is reported in the revised abstract. 7×10^{-9} mol/min/g_cat is many orders of magnitude lower than all catalysts listed in Table R2. While this isn't all that surprising since the other rates are at higher temperatures for alkanes, it should nonetheless be reported and calculated in a fair way.

Supplemental information should also include all of the information used to perform the calculations in the table (e.g. inlet flows, outlet flows detected, reactor volume, catalyst mass, surface area, etc.) so that reproduction of the values is possible.

Re: We thank the reviewer for the comments and being supportive. The efficiency of our reaction system was recalculated and updated in Table R1 as suggested.

According to the scald-up experiment, a H₂ production rate of 1.2×10^{-4} mol/min was obtained by using 5.0 g of SnIn_{0.1034}Cu_{0.0094} droplets as catalytic materials. Hence, the production rate for per gram catalyst in our case was calculated to be 2.4×10^{-5} mol·min⁻¹·g_{catalyst}⁻¹. The reaction processes, including inlet flows and outlet flows analysis, were included in Method section, and surface area calculations were presented in Supplementary discussions. Additionally, the different reaction conditions, such as catalyst mass, reactor volume, reaction temperature, reaction time and type of hydrocarbon sources, were included in Supplementary tables.

Table R1 in the next page was included in Supplementary information as Supplementary Table 10.

Table R1 | Comparison with reported technologies for dehydrogenation reactions

Catalytic materials	Feedstock	Reaction temperature (°C)	Pressure	Reaction types	Selectivity (%)	Efficiency (mol·min ⁻¹ ·g _{catalyst} ⁻¹) ⁽¹⁾	Ref.
Pt–Sn/CeO ₂	Propane	680	n/a	Dehydrogenation reaction	84.5	~8.0×10 ⁻²	4
[PtZn ₄]	Propane	520-620	n/a	Dehydrogenation reaction	< 95.0	~1.3	5
PtGa-Pb	Propane	600	n/a	Dehydrogenation reaction	99.6	~8.0×10 ⁻³	6
Pt-Sn/Mg-Al	Methylcyclohexane	300	n/a	Dehydrogenation reaction	~90.5	~5.0×10 ⁻³	7
Pt/TiO ₂	Methylcyclohexane	400	n/a	Dehydrogenation reaction	99.9	~2.0×10 ⁻²	8
Ga ₅₂ Pt/SiO ₂	Methylcyclohexane	450	1 bar	Dehydrogenation reaction	85.0	~7.2×10 ⁻⁵	9
Ga-Pd	Butane	445-500	1.1 bar	Dehydrogenation reaction	85.0	n/a	10
Ga–Rh	Propane	550	n/a	Dehydrogenation reaction	92.0	~5.8×10 ⁻³	11
Ga ₈₄ Pt/Al ₂ O ₃	n-heptane	410-470	n/a	Dehydrogenation reaction	~80.0	~4.5×10 ⁻⁴	12
SnIn _{0.1034} Cu _{0.0094}	Canola oil	260	n/a	Dehydrogenation reaction	~93.0	~2.4×10 ⁻⁵ ⁽²⁾	This work

Notes: (1) The efficiencies of the catalysts were calculated based on the weight of functional materials without including the weight of supporting species. (2) The rate was calculated based on the scald-up experiment by using 5.0 g of SnIn_{0.1034}Cu_{0.0094} droplets (not just the surface) as catalytic materials. Considering only the surface area, the estimated production rate is ~9.0×10⁻³ mol·min⁻¹·g_{catalyst}⁻¹.

Reviewer #3 (Remarks to the Author):

The work can be published.

Re: We thank the reviewer for being supportive and greatly appreciate the time and effort dedicated to improving our manuscript.

Reviewer #4 (Remarks to the Author):

The envisaged corrections have been made and mostly the comments have been addressed. In the reply, the authors clearly acknowledge the shortcomings of the computational model. However, this has not been added to the main text of the manuscript. I will be very supportive of this work if these are added to the main text along with the effects they can have on the conclusions of this work.

Re: We thank the reviewer for the comments and being supportive. The related discussions “Originating from the dynamicity and mobility of liquid atoms, AIMD simulations face limitations for molten alloys. The simulation boxes of liquid metallic atoms are small and periodic, and the timescales are short. Moreover, the high computational cost of modelling liquid hexadecane interfaced with liquid metal restricts the simulation to one single hexadecane molecule.” were included in the main manuscript on page 12.

References:

1. Song H. *et al.* Ga-based liquid metal micro/nanoparticles: recent advances and applications. *Small* **16**, 1903391 (2020).
2. Um H. J., Kong, G. D., Yoon, H. J. Thermally controlled phase transition of low-melting electrode for wetting-based spontaneous top contact in molecular tunnel junction. *ACS Appl. Mater. Interfaces* **10**, 34758-34764 (2018).
3. Batzill M., Diebold, U. The surface and materials science of tin oxide. *Prog. Surf. Sci.* **79**, 47-154 (2005).
4. Xiong H. *et al.* Thermally stable and regenerable platinum–tin clusters for propane dehydrogenation prepared by atom trapping on ceria. *Angew. Chem. Int. Ed.* **56**, 8986-8991 (2017).
5. Chen S. *et al.* Propane dehydrogenation on single-site [PtZn₄] intermetallic catalysts. *Chem* **7**, 387-405 (2021).
6. Nakaya Y., Hirayama, J., Yamazoe, S., Shimizu, K.-i., Furukawa, S. Single-atom Pt in intermetallics as an ultrastable and selective catalyst for propane dehydrogenation. *Nat. Commun.* **11**, 2838 (2020).
7. Yan J. *et al.* Dehydrogenation of methylcyclohexane over PtSn supported on MgAl mixed metal oxides derived from layered double hydroxides. *Int. J. Hydrog. Energy* **43**, 9343-9352 (2018).
8. Yang X. *et al.* The double tuning effect of TiO₂ on Pt catalyzed dehydrogenation of methylcyclohexane. *Mol. Catal.* **492**, 110971 (2020).
9. Sebastian O. *et al.* Stable and selective dehydrogenation of methylcyclohexane using supported catalytically active liquid metal solutions – Ga₅₂Pt/SiO₂ SCALMS. *ChemCatChem* **12**, 4533-4537 (2020).
10. Taccardi N. *et al.* Gallium-rich Pd–Ga phases as supported liquid metal catalysts. *Nat. Chem.* **9**, 862-867 (2017).
11. Raman N. *et al.* Highly effective propane dehydrogenation using Ga–Rh supported catalytically active liquid metal solutions. *ACS Catal.* **9**, 9499-9507 (2019).
12. Sebastian O., Al-Shaibani, A., Taccardi, N., Haumann, M., Wasserscheid, P. Kinetics of dehydrogenation of n-heptane over GaPt supported catalytically active liquid metal solutions (SCALMS). *React. Chem. Eng.* **9**, 1154-1163 (2024).